

# Mean string field theory:
# Landau-Ginzburg theory for 1-form symmetries

**Nabil Iqbal[1] and John McGreevy[2]**

**1** Centre for Particle Theory, Department of Mathematical Sciences,
Durham University, South Road, Durham DH1 3LE, UK
**2** Department of Physics, University of California at San Diego,
La Jolla, CA 92093, USA

## Abstract

By analogy with the Landau-Ginzburg theory of ordinary zero-form symmetries, we introduce and develop a Landau-Ginzburg theory of one-form global symmetries, which we call *mean string field theory*. The basic dynamical variable is a string field – defined on the space of closed loops – that can be used to describe the creation, annihilation, and condensation of effective strings. Like its zero-form cousin, the mean string field theory provides a useful picture of the phase diagram of broken and unbroken phases. We provide a transparent derivation of the area law for charged line operators in the unbroken phase and describe the dynamics of gapless Goldstone modes in the broken phase. The framework also provides a theory of topological defects of the broken phase and a description of the phase transition that should be valid above an upper critical dimension, which we discuss. We also discuss general consequences of emergent one-form symmetries at zero and finite temperature.

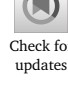

# 1  Introduction

This paper is concerned with an extension of the conventional Landau paradigm.

The Landau paradigm is one of the cornerstones of our understanding of nature [1]. In a nutshell, it states that phases of matter can be understood in terms of the global symmetries that they spontaneously break. Furthermore, continuous transitions between different phases are described by universal critical theories that are independent of microscopic details, depending only on symmetries and their patterns of breaking.

Of course, much fascinating work in modern physics is devoted to precisely those phases (and transitions) that are not accommodated by the usual Landau paradigm. Well-known examples (to which we devote the most attention) are phases whose low-energy description involves an emergent deconfined gauge theory; such systems are usually said to exhibit *topological order*. These phases break no ordinary symmetries and possess no local order parameters, and so clearly do not fit into the usual Landau classification [2–4].

It is thus interesting to note that we have recently learned of a generalization of the concept of "symmetry". The idea behind *higher-form symmetries* is simple [5]: just as ordinary global symmetries result in conservation laws for particles, theories that are invariant under higher-form global symmetries possess conservation laws for extended objects, such as strings or flux tubes. Importantly, these higher-form symmetries are on precisely the same conceptual footing as ordinary global symmetries. We are thus led to consider an *enlarged Landau paradigm*, where phases of matter are classified by their realization of higher-form symmetries. This addition to the toolkit of Landau dramatically expands the set of systems that he can describe, essentially providing a global symmetry formulation of features normally ascribed to a gauge-

theoretical description. Fascinatingly, many examples of topological order can be understood in this manner as spontaneously breaking (emergent) discrete higher-form symmetries. (The program of understanding topological order from a generalization of the notion of symmetry was initiated in [6,7]. Other early work on generalized symmetries is described in [8].)

For concreteness, we briefly review the story for a theory that is invariant under a $U(1)$ $p$-form symmetry. This possesses a conserved current that has $p + 1$ antisymmetric indices. The usual case of $p = 0$ is a conserved density of particles, with an ordinary particle-number current $j^\mu$. In this work we will focus on the case $p = 1$; we then have a 2-index current $J^{\mu\nu}$ which can count a density of conserved strings. The charged objects under this symmetry are *line operators* $W(C)$, where $C$ is a closed 1-dimensional curve. These line operators can be regarded as order parameters for symmetry breaking: if the symmetry is unbroken, $W(C)$ obeys an area law:

$$\langle W(C) \rangle \sim \exp(-\alpha \, \text{Area}[C]), \tag{1}$$

decaying with the area of the minimal-area surface bounded by the curve $C$, and where $\alpha$ can be understood as a string tension. On the other hand, if the symmetry is spontaneously broken, then $W(C)$ instead obeys a perimeter law:

$$\langle W(C) \rangle \sim \exp(-\beta \, \text{Perimeter}[C]). \tag{2}$$

An example of a theory enjoying such a 1-form symmetry is Maxwell electrodynamics coupled to electrically charged matter; the conserved current is precisely magnetic flux density $J^{\mu\nu} = \frac{1}{2}\epsilon^{\mu\nu\rho\sigma}F_{\rho\sigma}$. The charged line operator $W(C)$ is a 't-Hooft line, and the unbroken and spontaneously broken phases correspond to the Higgs and Coulomb phase of electromagnetism respectively [5,9]. Another familiar example is pure $SU(N)$ gauge theory; the "center symmetry" can be understood as a $\mathbb{Z}_N$ 1-form symmetry under which the fundamental Wilson line is charged. The unbroken and spontaneously broken phases of the 1-form symmetry correspond to the confined and deconfined phases of the gauge theory respectively.

We turn now to the second tenet of the Landau paradigm: the existence of universal Landau-Ginzburg theories that describe phase transitions in terms of the dynamics of a coarse-grained order parameter. It is clearly of interest to ask if a similar framework exists for higher-form symmetries. At first glance, this problem may seem rather daunting; after all, usual Landau-Ginzburg theories involve the condensation of *particles*. To build a similar theory for 1-form symmetries, it appears we need a framework to describe non-perturbatively the creation, destruction and condensation of *strings*. At least in the context of fundamental string theory, this is a famously difficult problem.

In this work we nevertheless build such a description by harnessing the constraints of higher-form symmetry; we dub the resulting object *mean string field theory*. The basic degree of freedom is no longer a field but instead a *functional* of closed curves $\psi[C]$, and it may be thought of as a representation of the charged line operator $W(C)$ defined above. To anchor the discussion and anticipate what follows, we write down the action for our theory here:

$$S[\psi] = \mathcal{N} \int [dC] \left( \frac{1}{2L[C]} \oint_C ds \, \frac{\delta \psi^\dagger[C]}{\delta \sigma_{\mu\nu}(s)} \frac{\delta \psi[C]}{\delta \sigma^{\mu\nu}(s)} + V(\psi^\dagger[C]\psi[C]) \right) + S_{\text{tc}}[\psi]. \tag{3}$$

Here $[dC]$ is a functional integral over closed curves, and $\frac{\delta}{\delta\sigma^{\mu\nu}}$ is the "area derivative" familiar from the loop formulation of non-Abelian gauge theory [10–12]. $S_{\text{tc}}$ includes topology-changing terms that we discuss later.

In the remainder of this paper we present a detailed construction of this action and a study of some of its consequences. We will see that this allows a straightforward and physically transparent understanding of many features of higher-form symmetry and its breaking. In

particular, we will derive the existence of an area law in the unbroken phase and directly understand the dynamics of emergent Goldstone modes in the spontaneously broken phase. We will also discuss the possibility of observing the mean-field phase transition and its associated scaling exponents for quantities such as the string tension.

It is important to note that (like conventional Landau-Ginzburg theory), this is not a UV-complete description; instead we presume it might describe long-distance dynamics near a phase transition. Our goals are therefore substantially less ambitious – and the structure of our action rather different — than those of traditional *fundamental* string field theory [13, 14], which seeks to non-perturbatively describe the theory of fundamental strings (and thus quantum gravity).

**Connections to previous work.** We hasten to note that actions similar to (3) have been studied before: in particular [15] has studied a very similar action coupled to a dynamical 2-form field, motivated by a higher-form version of the Higgs mechanism for the Kalb-Ramond 2-form field in fundamental string theory. That formalism has been applied to the theory of superfluids (systems with a global 0-form $U(1)$ symmetry) in $D = 3 + 1$ in [16, 17], where the 0-form symmetry is associated with the topological current of the 2-form gauge field. In contrast, our focus is on a system where the 1-form symmetry is not gauged, and there is no dynamical gauge field[1] associated with the 2-form current. We are concerned only with the global symmetries, which indeed largely dictate the structure of the theory.

Previous work has attempted to reformulate gauge theory as a theory of loops [10–12, 18, 19]. In fact, [18, 19] derived a theory of the form of (1.3) on the lattice starting from lattice gauge theory by path integral manipulations. This construction even allows for matrix-valued string fields. Yoneya's paper [19] in particular notes the transformation that is in modern language a 1-form symmetry, as well as the emergence of a photon in the broken phase. Our symmetry-based perspective is complementary to this interesting work; here instead, we construct the generic theory that realizes linearly a 1-form symmetry (this of course includes certain gauge theories) and study its behavior.

**Plan.** An outline of the rest of the paper follows. In Section 2 we provide a lightning overview of 0-form conventional Landau-Ginzburg theory to set the stage, and then describe the machinery and symmetry principles used to construct the mean string field theory action for a $U(1)$ 1-form symmetry. In Section 3 we discuss the theory in its unbroken phase, and demonstrate that the string field satisfies an area law for large curves. In Section 4 we study the theory in the spontaneously broken phase, and demonstrate the existence of a Goldstone mode that can be thought of as an emergent gauge field. We also initiate a classification of topological defects of the resulting ordered medium. In Section 5 we discuss the generalization of the machinery to the case of a discrete 1-form symmetry, showing that the spontaneously broken phase is now described by the expected topological quantum field theory. Section 6 addresses a crucial question about when we can expect Mean String Field Theory to be useful: here we discuss the consequences of emergent 1-form symmetries, and the generic irrelevance of symmetry breaking deformations, in contrast to the case of 0-form symmetries. We present some of our arguments in a more general language that does not require the string field machinery. Finally, we conclude in Section 7 with some possible applications, caveats of our analysis, and directions for future research. Various calculational details and formal manipulations are relegated to the appendices.

---

[1]Indeed, it appears to be something of a historical accident that the formalism for the *gauged* versions of higher-form symmetries – which are ubiqutious in supergravity and string theory – is far more developed than the same formalism for the conceptually simpler global higher-form symmetries.

## 2 String field action

### 2.1 Conventional mean field theory

We begin with a short review of the ideas that go into the textbook construction of a Landau-Ginzburg theory. This is merely to provide an explicit template for the generalization that follows, and the impatient reader can skip straight to the next section. Let us imagine that we would like to describe the universal properties of a phase transition in which a (conventional 0-form, global) $U(1)$ symmetry is spontaneously broken. How do we do this?

The first step is to imagine the existence of a continuous field $\phi(x)$ that is charged under the global $U(1)$ symmetry, i.e. that transforms linearly as

$$\phi(x) \to e^{iq\alpha}\phi(x), \qquad d\alpha = 0. \tag{4}$$

We note the trivial fact that $\phi(x)$ is a map from spacetime points to $\mathbb{C}$. This field is usually thought of as a coarse-grained representation of microscopic degrees of freedom, and a description in terms of $\phi$ is valid below some scale $\Lambda$. As usual, in field theory $\phi(x)$ is interpreted as creating a particle at the spacetime point $x$, and the $U(1)$ charge counts these particles.

It is often helpful to imagine coupling the $U(1)$ symmetry to a fixed *external* gauge field $\mathcal{A}_\mu$. This coupling is completely determined by the demand that the coupled system be invariant under the following enlarged version of (4):

$$\phi(x) \to e^{iq\alpha(x)}\phi(x), \qquad \mathcal{A}(x) \to \mathcal{A}(x) + d\alpha(x), \tag{5}$$

where now $\alpha(x)$ is a function of spacetime.

We now proceed to write down the most general Euclidean action that is (a) invariant under the symmetry above, and (b) is local in spacetime. This takes the form:

$$S[\phi; \mathcal{A}] = \int d^d x \left( (D_\mu \phi)^\dagger (D^\mu \phi) + m^2 \phi^\dagger \phi + \frac{\lambda}{4} \phi^\dagger \phi + \cdots \right), \tag{6}$$

where $D_\mu$ is the gauge-covariant derivative invariant under (5):

$$D_\mu \phi = \partial_\mu \phi - iq\mathcal{A}_\mu \phi. \tag{7}$$

We can obtain the current $j^\mu$ from the action by functional differentiation with respect to the source $\mathcal{A}_\mu$: $j^\mu(x) = \frac{\delta S}{\delta \mathcal{A}_\mu(x)}$.

The action (6) describes two phases, depending on the sign of $m^2$. If $m^2 > 0$, then the system is gapped, with a unique minimum for the potential at $\phi = 0$. The correlation function of $\phi(x)$ decays exponentially:

$$\langle \phi^\dagger(x)\phi(y) \rangle \sim e^{-\frac{|x-y|}{\xi}}, \tag{8}$$

with $\xi = m^{-1}$. This is the phase where the symmetry is *unbroken*.

On the other hand, if $m^2 < 0$, then the potential will have a non-trivial minimum at some nonzero value of $|\phi| = v$. However due to the $U(1)$ symmetry there will be a light Goldstone mode $\theta(x)$, in terms of which we can parametrize $\phi(x)$ as

$$\phi(x) = v e^{i\theta(x)}. \tag{9}$$

Expanding the action we find

$$S[\theta; \mathcal{A}] = \int d^d x \left( v^2 (d\theta - q\mathcal{A})^2 + \cdots \right). \tag{10}$$

This shows clearly the existence of a gapless Goldstone mode $\theta(x)$. This is the phase where the symmetry is *spontaneously broken*. Note that at long distances the two point function of $\phi$ factorizes and becomes independent of the relative separation, but is still nonzero, unlike in the unbroken phase:

$$\lim_{|x-y|\to\infty} \langle \phi^\dagger(x)\phi(y)\rangle \sim \langle \phi^\dagger(x)\rangle\langle\phi(y)\rangle \sim v^2 \,. \tag{11}$$

This may be viewed as an order parameter for the spontaneous breaking of the symmetry.

Finally, at $m^2 = 0$ we find a continuous transition between these two phases. This is described by a conformal field theory. Whether or not this CFT can be described reliably by the action above depends on the value of the spacetime dimension $d$: if $d$ is greater or equal to the upper critical dimension (which for the $U(1)$ case above is 4), then interactions are irrelevant at the transition, and the action above provides quantitatively correct answers for critical exponents; for example one can interpret (8) as saying

$$\xi \sim |m^2 - m_c^2|^{-\frac{1}{2}} \,, \tag{12}$$

with $m_c^2 = 0$, i.e. the correlation length diverges with a square root exponent at the critical point.

If $d$ is less than the upper critical dimension, then interactions cannot be ignored and we will flow to a strongly interacting fixed point with non-trivial critical exponents. Nevertheless, the action above is still helpful in describing the structure of the phase diagram and the gross character of the phases on either side of the transition, though care must be taken at the critical point itself. The action above is also useful as a starting point for an $\epsilon$-expansion about the upper critical dimension.

## 2.2 String field

We now provide an analogous construction for a 1-form symmetry.

The first step is to identify the correct degrees of freedom. Just as local operators are charged under 0-form symmetries, *line operators* are charged under 1-form symmetries.

We are thus led to postulate that the role of the order parameter field $\phi(x)$ is now played by a field $\psi[C]$, where $C$ is a closed connected curve in spacetime. $\psi[C]$ is a map from the space of closed curves to $\mathbb{C}$, and is thus not a function but a functional. We will call $\psi[C]$ the *string field*. This functional will be the main dynamical degree of freedom in our theory; it clearly contains much more information than in the 0-form case, and much of what follows will revolve around extracting physical observables from the string field.

Under the global $U(1)$ 1-form symmetry it transforms linearly, as

$$\psi[C] \to \psi[C]e^{iq\int_C \Lambda}, \qquad d\Lambda = 0 \,, \tag{13}$$

where $\Lambda$ is a closed 1-form that plays the role of the symmetry transformation parameter $\alpha$ in (4). We imagine that $\psi[C]$ creates a string living along $C$, and the integrated 1-form charge counts these strings. Note that the curves $C$ must be closed loops for invariance under the 1-form symmetry, and thus our framework only involves closed strings.

On a trivial spacetime topology, the closedness condition means that there are no $\Lambda$ that contribute nontrivially to the above transformation of $\psi[C]$.[2] The expression above may thus seem to have less dynamical content than the corresponding expression (4) in the 0-form case. To elevate it to an organizing principle, it is helpful to consider coupling this $U(1)$ symmetry to

---

[2]This encodes the idea that a closed string that doesn't wrap anything may shrink to zero and vanish.

an external 2-form source $\mathcal{B}_{\mu\nu}$, analogous to (5). The coupling is determined by the following enlarged symmetry operation

$$\psi[C] \to \psi[C]e^{iq\int_C \Lambda}, \qquad \mathcal{B}(x) \to \mathcal{B} + d\Lambda, \tag{14}$$

where now $\Lambda$ is an *arbitrary* 1-form.

We emphasize that as in the case of ordinary Landau-Ginzburg theory, we expect that $\psi[C]$ is a coarse-grained variable, and that the description in terms of $\psi$ is only valid below some cutoff scale. Next we construct an action for the field $\psi[C]$ that is invariant under (14).

## 2.3 Area derivative

Our action will require a kinetic term, and thus a necessary first step is to understand how to take a "derivative" of the string field $\psi[C]$. The technology that we will require – that of the *area derivative* – was developed long ago for the loop formulation of non-Abelian gauge theory in [10–12]; see also an application to fluid turbulence in [20]. For completeness, we provide a brief review of the ideas here, and refer the reader to the original work for a more detailed discussion.

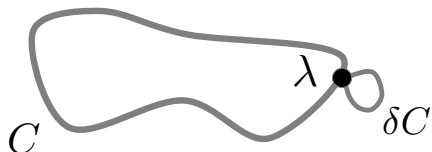

Figure 1: Adding a small loop $\delta C$ to the curve $C$ at $\lambda$ to take the area derivative.

In [10], the area derivative is defined as follows. Consider a functional $\psi[C]$, and a curve $C$ with a point $\lambda$ on that curve. Then imagine changing the curve $C$ by adding a new little closed loop $\delta C$ at the point $\lambda$, as in Figure 1. Then the functional is expected to shift, and this shift should depend only on invariant geometric properties of $\delta C$. In the limit that $\delta C$ is very small, we have:

$$\psi[C \cup \delta C] = \psi[C] + \sigma^{\mu\nu}(\delta C)\frac{\delta\psi[C]}{\delta\sigma^{\mu\nu}(\lambda)}. \tag{15}$$

This is now taken as the definition of the area derivative $\frac{\delta\psi[C]}{\delta\sigma^{\mu\nu}(\lambda)}$. Here $\sigma^{\mu\nu}(\lambda)$ is the infinitesimal bit of area formed by integrating the following over the curve $\delta C$:

$$\sigma^{\mu\nu}(\delta C) = \frac{1}{2}\oint_{\delta C} dX^\mu X^\nu. \tag{16}$$

Note that $\sigma_{\mu\nu}$ is antisymmetric; trying to construct its symmetric part we have

$$\oint_{\delta C} (dX^\mu X^\nu + dX^\nu X^\mu) = \oint_{\delta C} d(X^\mu X^\nu) = 0. \tag{17}$$

$\sigma^{\mu\nu}$ is the simplest geometric invariant associated with the little bit of curve $\delta C$; we thus see that the area derivative is naturally antisymmetric.

This definition allows us to compute the area derivatives of various functionals of the curve $C$. To demonstrate the approach, below we explicitly compute the area derivative of the following functional:

$$\phi[C] \equiv \oint_C dX^\mu \mathcal{A}_\mu, \tag{18}$$

where $\mathcal{A}_\nu$ is a fixed external gauge field. In particular, we see that if we deform the contour with a small $\delta C$ the difference is:

$$\phi[C \cup \delta C] - \phi[C] = \oint_{\delta C} dX^\mu \mathcal{A}_\mu(X) = \oint_{\delta C} dX^\mu \left( \mathcal{A}_\mu(X_0) + (X_0 - X)^\nu \partial_\nu \mathcal{A}_\mu(X_0) + \cdots \right), \quad (19)$$

where we expand in powers about $X_0 = X_C(\lambda)$, i.e. the point where the new curve is attached. Using the antisymmetry exhibited above, this becomes

$$-\frac{1}{2} \oint_{\delta C} dX^\mu X^\nu \cdot 2 \partial_{[\nu} \mathcal{A}_{\mu]}(X_0) = \sigma^{\mu\nu}(\delta C) \mathcal{F}_{\mu\nu}(X_0), \quad (20)$$

where $\mathcal{F} = d\mathcal{A}$. In other words, the area derivative of the coupling to the gauge field is equal to the field strength, evaluated at the point on the curve where we take the derivative [21]:

$$\frac{\delta \phi[C]}{\delta \sigma^{\mu\nu}(\lambda)} = \mathcal{F}_{\mu\nu}(X_C(\lambda)). \quad (21)$$

We may now turn to the construction of a gauge-covariant derivative of the string field, i.e. the 1-form analogue of (7). Using the relation above, we see that the following 1-form gauge covariant derivative

$$\frac{D\psi[C]}{\delta \sigma^{\alpha\beta}(s)} = \left( \frac{\delta}{\delta \sigma^{\alpha\beta}(s)} - iq \mathcal{B}_{\alpha\beta}(x_C(s)) \right) \psi[C] \quad (22)$$

transforms covariantly under the 1-form symmetry transformation (14), i.e.

$$\frac{D\psi[C]}{\delta \sigma^{\alpha\beta}(s)} \rightarrow e^{iq \int_C \Lambda} \frac{D\psi[C]}{\delta \sigma^{\alpha\beta}(s)}, \quad (23)$$

with no inhomogenous terms. Note that the area derivative is precisely the correct object for this construction, and indeed this is why the formalism requires the area derivative rather than a more conventional functional derivative with respect to the coordinates of the curve $X^\mu(\lambda)$.[3] We can now use the derivative above to construct an invariant action. A key role in the analysis of the action is played by the area derivative of the minimal area functional, which we record here.

**Minimal area**: Consider the functional $A[C]$, which is defined only for contractible $C$, and is the area of the minimal surface that "fills in" $C$ (i.e. the area of the minimal 2d surface $\mathcal{M}$ such that $\partial \mathcal{M} = C$). Its area derivative is

$$\frac{\delta A[C]}{\delta \sigma^{\mu\nu}(\lambda)} = n_\mu t_\nu - n_\nu t_\mu, \quad (24)$$

where $n_\mu$ is the outwards pointing normal vector to the curve $C$ along the surface, $t_\mu$ is the normalized tangent vector along $C$, and where the right hand side is evaluated at the point $\lambda$ along the curve where we take the derivative. As the minimal area is a rather non-local concept, it may appear surprising that the answer above depends only locally on the data of the curve. Heuristically, this comes about because modifying the boundary condition on the minimal surface by altering $C$ has no effect on the interior of the surface (which is minimal, and thus stationary under small variations), and the variation comes only from a boundary term. The derivation of this result – along with a few other useful area derivatives – is reviewed in Appendix A (see also [10]).

---

[3]The area derivative can also be defined as a particular combination of functional derivatives: see [10], though this definition is not manifestly reparametrization invariant.

## 2.4   Mean string field theory

We are now ready to construct an action. We will impose the following conditions:

1.  Invariance under the 1-form global symmetry in the form: (14).

2.  Euclidean rotational invariance and translational invariance in the target space $\mathbb{R}^d$ in the usual form.

3.  Translational invariance in the space of curves; as explained in Appendix B, we demand that the action be invariant under (almost) all infinitesimal movements in loop space $\delta X^\mu(\lambda)$.

With no further ado, we propose the following action for *mean string field theory*:

$$S_0[\psi;\mathcal{B}] = \mathcal{N}\int [dC]\left(\frac{1}{2L[C]}\oint_C ds\,\frac{D\psi^\dagger[C]}{\delta\sigma_{\mu\nu}(s)}\frac{D\psi[C]}{\delta\sigma^{\mu\nu}(s)} + V(\psi^\dagger[C]\psi[C])\right). \tag{25}$$

The integral $ds$ in the kinetic term is taken with respect to the proper length $s$ along the curve; in terms of a more general parameter $\lambda$ we could write it as $ds = d\lambda\sqrt{\dot{X}^2}$, where an overdot denotes a $\lambda$ derivative. There are two types of interaction in this action. The potential we consider takes the usual form:

$$V(\psi^\dagger\psi) = r\psi^\dagger\psi + \frac{u}{4}(\psi^\dagger\psi)^2 + \cdots, \tag{26}$$

with $r,u$ constant couplings.

$\mathcal{N}$ is a normalization of the action. Much of our discussion will be classical, and $\mathcal{N}$ will not concern us until Section 4.

We note that this is actually not the most general action one can write down. Importantly, there is an entire class of interactions that we have omitted, which take the form:

$$S_\text{tc} = \lambda\int [dC_1][dC_2][dC_3]\delta[C_1-(C_2+C_3)]\psi^\dagger[C_1]\psi[C_2]\psi[C_3] + \text{h.c.} + \cdots. \tag{27}$$

The ingredients here deserve some explanation. The sum and difference over individual loops is taken in the sense of providing an oriented region of integration; e.g. see Figure 2 for an example of a configuration of loops in which $C_1 = C_2 + C_3$, and thus for which the delta function has support. The form of the delta function means that the term is still invariant under the 1-form symmetry (14); it represents a topology-changing interaction in which two strings merge to form a larger one. Though unfamiliar, such terms are allowed by the symmetries and should be considered. Indeed, if they were not present in the bare action it seems they would be generically produced by the $u|\psi|^4$ term in the action (combined with the boundary condition on small loops plus ordinary string propagation via the kinetic term) if fluctuations were allowed. (Note they have no simple analogue in the usual 0-form theory).

In the bulk of this paper we will nevertheless assume that such terms do not exist. To our knowledge this cannot be justified on symmetry grounds, and thus amounts to fine-tuning. We believe that including such terms would not significantly affect the physics deep in the various phases that we study. However, they may potentially have significant effects at the phase transition itself. We discuss this in the conclusion. In the remainder of the paper we simply study a model with such terms fine-tuned to zero, in order to permit a reasonably self-contained analysis.

A further fine-tuning arises from the form of the couplings such as $r$ and $u$. In particular, the symmetries described in Appendix B permit such couplings in the action to be arbitrary functions of the invariant length of the curve $L[C]$, though they forbid any other dependence.

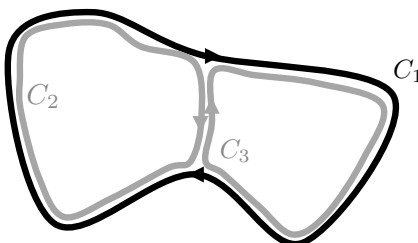

Figure 2: Example of three curves $C_{1,2,3}$ on which the delta function in loop space $\delta[C_1 - C_2 + C_3]$ has support.

In this work we *assume* they are constant; it would be extremely desirable to find a more restrictive symmetry principle that would enforce this while still permitting the kinetic term.

Having explained the caveats, we now return to the discussion of the action, which is the natural generalization of the familiar 0-form mean-field action described in (6). The definition of the action itself involves integrals over all closed connected curves $[dC]$. This is the analogue of the integral $d^d x$ for the 0-form case. We assume here that there is a factor that weights each curve by its length, i.e.

$$[dC] = [dX]e^{-mL[C]}, \tag{28}$$

where $[dX]$ is the usual functional integral over embeddings of a 1d curve in $\mathbb{R}^d$, and $m$ is a non-universal UV scale that suppresses long curves[4]. Our motivation here is that such a term generically arises when regularizing the integral over curves (see e.g. [22]), and setting it to zero would thus amount to fine-tuning.

As we will show below, the integral over curves can often be done using standard techniques from worldline formulations of quantum field theory. Nevertheless, this is a considerable technical complication over conventional field theory.

The classical action above should be used to define a quantum theory, whose partition function is

$$Z[\mathcal{B}] \equiv \int [\mathcal{D}\psi]\exp\left(-S[\psi;\mathcal{B}]\right). \tag{29}$$

The path integral here is over all string fields, i.e. all functionals. One can in principle use this to calculate observables, such as the expectation value of the string field $\langle\psi[C]\rangle$ or the correlation function of the two-form current $J^{\mu\nu}$. These are precisely the observables of interest in higher-form symmetry and its breaking. (29) is a formal expression, and it is not obvious that a measure can be constructed with all the requisite invariances. The analysis below will not depend on this assumption.

At this point the reader may be alarmed; the path integral over all functionals clearly involves far more degrees of freedom than in a conventional local many-body system. Roughly speaking this is because the stringy excitations we are describing have many vibrational modes, each of which presumably corresponds to one spacetime field, as usual in string theory. However it may then seem ludicrous that such a string theory could describe any realistic phase transition involving local degrees of freedom[5]. We argue below that the vast majority of the degrees of freedom in this system are gapped at a very high scale; there is typically a far smaller light subsector that captures the dynamics of interest in an interesting manner.

---

[4]We note that such a choice of measure can also be thought of as a kind of wave-function regularization for the string field $\psi[C]$.

[5]See previous discussion on this point in [23].

What are the phases described by the string field theory action (30)? Just as in regular field theory, a key role is played by the sign of $r$ in the potential (127). If $r \gg 0$, the potential has a unique minimum at $\psi = 0$, and the vacuum leaves the 1-form symmetry unbroken. If $r < 0$, the potential is minimized at some nonzero value of $\psi$. This will necessarily transform under the 1-form symmetry, which is then spontaneously broken. We then expect to find light Goldstone modes [5, 9, 24, 25].

## 3 Unbroken phase

We begin by studying the theory in the phase where $r \gg 0$. To start, we ignore string interactions, i.e. we study an action quadratic in the string field:

$$S[\psi] = \mathcal{N} \int [dC] \left( \frac{1}{2L[C]} \oint_C ds \frac{\delta \psi^\dagger[C]}{\delta \sigma_{\mu\nu}(s)} \frac{\delta \psi[C]}{\delta \sigma^{\mu\nu}(s)} + r\psi[C]^\dagger \psi[C] \right). \tag{30}$$

We set the source $b$ to 0 in this section. We would now like to understand the solutions to the linear equations of motion arising from this action. In usual field theory the only translationally invariant solution is $\phi(x) = \text{const}$, and then the constant must vanish in the unbroken phase. Here we will see that it is possible to have other non-zero solutions to $\psi[C]$; in particular we will find an *area law* solution.

### 3.1 Area law

We will begin by varying the action to obtain the classical linear equations of motion. To do this, we need to integrate the action by parts with respect to the area derivative[6]. Here it is important to recall that the integration measure $[dC] = [dX] \exp(-mL[C])$ contains a factor which depends on the proper length; we find:

$$S[\psi] = \mathcal{N} \int [dX] \left( -\frac{1}{2} \oint_C ds \psi^\dagger[C] \frac{\delta}{\delta \sigma_{\mu\nu}(s)} \left( L[C]^{-1} e^{-mL[C]} \frac{\delta \psi[C]}{\delta \sigma^{\mu\nu}(s)} \right) \right.$$
$$\left. + r e^{-mL[C]} \psi[C]^\dagger \psi[C] \right). \tag{31}$$

Varying this with respect to $\psi[C]$, we may now write down the equations of motion directly in the space of curves:

$$-\frac{1}{2} e^{mL[C]} \oint_C ds \frac{\delta}{\delta \sigma_{\mu\nu}(s)} \left( ds \frac{e^{-mL[C]}}{L[C]} \frac{\delta \psi[C]}{\delta \sigma^{\mu\nu}(s)} \right) + r\psi[C] = 0. \tag{32}$$

This is an intriguing and somewhat formal equation; it is a differential equation in loop space. Given a particular way to parametrize the space of loops, it can be thought of as a partial differential equation in infinitely many variables. We have not been able to solve it in full generality; instead here we will discuss only the solution for *large* curves $C$.

To begin, let us consider the following ansatz for $\psi[C]$:

$$\psi[C] = \exp(\mathcal{S}(A[C])). \tag{33}$$

---

[6]The simplest way to justify such an integration by parts is by using the (not manifestly reparametrization invariant) expression for the area derivative in terms of ordinary functional derivatives given in [10], Eq. (2.56), which can be integrated by parts inside a functional integral.

Here $A[C]$ is the area of the minimal surface $\mathcal{M}$ such that $\partial \mathcal{M} = C$; in other words, we have assumed the string field depends only on the *area* that fills in $C$. This is a very particular dependence on the infinitely large set of data characterizing $C$. The choice is motivated by physical expectations, as well as by the simple form of the area derivative when acting on this functional; in particular, from (24) we have:

$$\frac{\delta \psi[C]}{\delta \sigma^{\mu\nu}(s)} = 2\mathcal{S}'(A) n_{[\mu} t_{\nu]} \psi[C]. \tag{34}$$

Anticipating the WKB-type analysis that will follow, we have further parametrized this dependence on $A$ in terms of a function of one variable $\mathcal{S}(A)$ in the exponent.

Inserting this ansatz into (32), we find

$$\mathcal{S}'(A)^2 + e^{mL[C]} \oint_C \frac{\delta}{\delta \sigma_{\mu\nu}(s)} \left( ds \frac{e^{-mL[C]}}{L[C]} n_{[\mu} t_{\nu]} \mathcal{S}'(A) \right) - r = 0. \tag{35}$$

In particular, note that the kinematic factors of normal and tangent vectors square to a constant which can be trivially integrated over the curve in the first term, which we have isolated as the only term quadratic in $\mathcal{S}(A)$.

The second term – linear in $\mathcal{S}(A)$ – here is intricate. One can use the expression for the area derivative of the length in (A.150) to work it out more explicitly; as we highlight in Appendix D, the resulting expression is interestingly singular, where coincident area derivatives integrated along the worldline result in an effective renormalization of the bare world-line tension $m$. More importantly, the kinematic factors no longer square to a constant; instead the answer depends on the full geometry of the curve and not just its area $A[C]$. Thus, for *generic* curves $C$ the area ansatz (33) does not apply.

However, let us now consider *large* curves $C$. On dimensional grounds, (assuming that the curve is "generic", in that all length scales characterizing it scale like $\sqrt{A}$) the second term scales at large $A$ at worst as $\frac{m}{\sqrt{A}} \mathcal{S}'(A)$. If we self-consistently ignore it at large areas, we then find the very simple WKB solution for $\mathcal{S}(A)$:

$$\mathcal{S}(A) = \pm\sqrt{r} A \left( 1 + \mathcal{O}(A^{-\frac{1}{2}}) \right). \tag{36}$$

Taking the solution that decays at large area and plugging back into $\psi[C]$, we find the following form for $\psi[C]$:

$$\psi[C] = c \exp\left( -\sqrt{r} A[C] \right). \tag{37}$$

This is the answer for the string field at large curves, which does indeed depend only on the minimal area. $c$ is an overall constant that is not fixed by our linearized analysis. $\sqrt{r}$ plays the role of the string tension.

We emphasize the appearance of the celebrated *area law*. This is usually taken to be a signature of confinement in non-Abelian gauge theory. It is however well-understood that the area law is a clean order parameter only if the theory in question has a (usually discrete) 1-form symmetry [5]. This is particularly transparent in our formalism, where the above behavior of the string field is indeed a robust consequence of the existence of an (unbroken) 1-form symmetry. Indeed, as the simplicity of the derivation makes clear, this expression is precisely analogous to the exponential decay of a correlator of local operators in (8).

Note that in the context of non-Abelian gauge theory, [10] reports a similar self-consistent area law solution of the *loop equation*. This equation – which imposes a certain gauge theory Ward identity – is distinct from our equations of motion. From the point of view of our theory, the loop equation seems to have the status of a condition for integrability.

This relation (37) can be viewed as the determination of a critical exponent. There is a phase transition at $r = 0$; the string tension in the area law should vanish at that point, and the

critical exponent controlling its approach to zero should be universal. As the control parameter is $r$, we may read off from above that the exponent is the usual mean-field value of $\frac{1}{2}$, i.e.

$$\text{tension} \sim |r - r_c|^{\frac{1}{2}}, \tag{38}$$

with $r_c = 0$. We also note that it is not obvious at the moment that the transition is necessarily second-order, and for this to happen may require fine-tuning. In the final section we compare this to lattice simulations and previous expectations.

It remains to fix the value of the overall constant $c$. We take the point of view that this should be determined by matching the effective string field theory to a microscopic description for small curves, i.e.

$$\psi[\text{infinitesimal loop}] = c, \tag{39}$$

with $c$ fixed by microscopics to some (presumably nonzero) value. The area law (37) is the main result of this section.

To see the need for a condition such as (39), we note that integrating the equations of motion requires a boundary condition at small loops. To see this, think of the equation of motion as a recursion relation determining the value of $\psi$ on a given loop from its value on smaller loops.

The boundary condition (39) (a) is consistent with the symmetries, since for small, contractible loops, $C = \partial R$, $\psi[C] \to e^{i \oint_C \Gamma} \psi[C] = e^{i \int_R d\Gamma} \psi[C] = \psi[C]$ is neutral, and (b) will match nicely to gauge theory in the broken phase. In particular, this amplitude $c$ can be matched to a gauge coupling (at the scale of the small loop) in the broken phase [10]. Physically, this boundary condition says simply that a small loop can shrink to nothing, with some fixed amplitude. In [26] we will show that such a boundary condition indeed arises from a lattice regularization of the mean string field theory.

## 3.2 Truncated action

It would be nice to have an understanding of the classical solutions away from the large $A$ limit. To that end, here we discuss another approach; we consider a *truncated action* where we brutally assume that the string field depends only on $A$. As argued above, this is actually not a justifiable ansatz for a particular choice of curve $C$; rather one could imagine that here we define a "coarse-grained" string field where we average $\psi[C]$ over all $C$ with a given $A$, and then consider the dynamics of this new effective field. This somewhat obscure interpretation of the effective string field means that this is not strictly a controlled approach; however we find that it provides a useful verification of the scaling of subleading terms displayed above, and we suspect it provides qualitatively correct answers.

We thus assume that $\psi[C]$ takes the form

$$\psi[C] = f(A[C]), \tag{40}$$

where $C$ is a contractible loop, where $A[C]$ is the area of the minimal surface that fills in the loop, and $f$ is a function of one variable to be determined. The area derivative acts on a simple way on this ansatz; in particular from (24) we have:

$$\frac{\delta \psi[C]}{\delta \sigma^{\mu\nu}(s)} = 2f'(A) n_{[\mu} t_{\nu]}. \tag{41}$$

Plugging this form into the action, we see that the kinematic factors in the kinetic term square to a constant, leaving:

$$S[f] = \mathcal{N} \int [dC] \left( f'(A[C])^2 \frac{1}{L[C]} \oint ds + V(f(A[C])) \right). \tag{42}$$

Doing the one-dimensional integral over the curve in the kinetic term, the reduced action simply becomes

$$S[f] = \mathcal{N} \int [dC] \big( f'(A[C])^2 + V(f(A[C])) \big). \tag{43}$$

(We note that this simple self-consistent form follows from the judicious choice of ansatz in (40)).

To proceed, we would like to reduce the measure to an integral only over the filled-in-area $A[C]$ of each curve. Formally we insert 1 into the integral in the following form:

$$1 = \int da\, \delta(a - A[C]). \tag{44}$$

We then find the following form for the reduced action

$$S[f] = \mathcal{N} \int da\, g(a) \big( f'(a)^2 + V(f(a)) \big), \tag{45}$$

where $g(a)$ is a measure over areas that is inherited from the integral over all curves

$$g(a) = \int [dC]\, \delta(a - A[C]). \tag{46}$$

Given a form for $g(a)$, extremizing the action in the form (45) to determine the value of the string field $f(a)$ is a simple exercise in classical mechanics. Furthermore, computing the density of areas $g(a)$ from (46) is a well-posed problem in worldline quantum field theory. It seems somewhat difficult to obtain a general expression, but we can obtain the large $a$ behavior through a saddle-point approximation of the worldline path integral. This is done in Appendix C, and the answer is

$$g(a \to \infty) \sim g_0 \exp\left( -2m\sqrt{\pi a} \right), \tag{47}$$

where $m$ is the worldline tension defined in (28). The saddle in question is a circle of area $a$, and the density of states is suppressed by a factor depending on the perimeter of this circle.

Inserting this into (45), we find the new equation of motion, which should hold for $a \gg m^{-2}$:

$$f''(a) - \frac{m\sqrt{\pi}}{\sqrt{a}} f'(a) - r f(a) = 0. \tag{48}$$

This ODE does not admit an analytic solution, though it can be easily solved numerically. We can study the large $a$ limit analytically. As above, it is helpful to consider a WKB ansatz $f(a) = \exp(\mathcal{S}(a))$; plugging this into (48) we find the equation

$$\mathcal{S}''(a) + \mathcal{S}'(a)^2 - m\sqrt{\frac{\pi}{a}} \mathcal{S}'(a) - r = 0. \tag{49}$$

Let us neglect the term $\mathcal{S}''(a)$. We can then explicitly solve for $\mathcal{S}(a)$. The solution that dies off at infinity is

$$\mathcal{S}(a) = \frac{1}{2}\sqrt{a}\left( 2m\sqrt{\pi} - \sqrt{m^2\pi + 4ar} \right) - \frac{m^2\pi}{4\sqrt{r}} \sinh^{-1}\left( \frac{2\sqrt{ar}}{m\sqrt{\pi}} \right) + \text{const}. \tag{50}$$

Comparing this to the exact numerical solution of the ODE in Figure 3 we find excellent agreement at large $a$; indeed we can verify that on this solution $\mathcal{S}''(a) \sim a^{-\frac{3}{2}}$, self-consistently justifying our neglect of this term at large $a$.

Expanding $\mathcal{S}(a)$ at large $a$ we find

$$\mathcal{S}(a \to \infty) \approx -\sqrt{r}a + m\sqrt{\pi a} - \frac{m^2\pi}{8\sqrt{r}}\left(1 + \log\left(\frac{16ar}{m^2\pi}\right)\right) + \mathcal{O}(a^{-1}). \tag{51}$$

The dominant term is the expected area law argued for above, with mean-field exponent for the string tension. We also see a subleading dependence that scales roughly as a linear length scale $\sqrt{a}$ characterizing the curve (weighted by $m$); for a smooth curve this scales the same way as a perimeter law dependence.

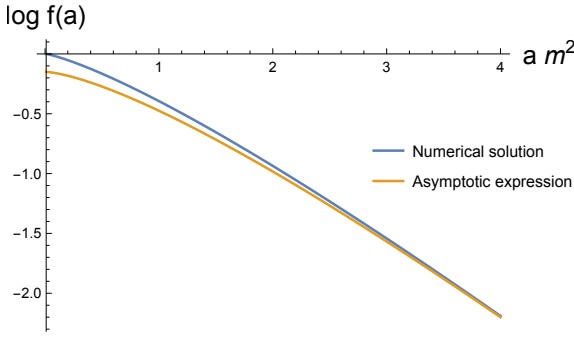

Figure 3: Comparison of numerical solution of ODE (48) with WKB asymptotic approximation (50) for $rm^{-4} = 1$. Integration constant in the asymptotic expression has been adjusted for agreement at large $a$.

To conclude, we note that it should be possible to do better than we have done so far. Let us briefly sketch how we might solve the infinite-dimensional PDE (32); we first form a basis of geometric invariants describing the curve $C$ as

$$T^{\mu_1\mu_2\cdots\mu_N} \equiv \oint_C dsX^{\mu_1}X^{\mu_2}\cdots dX^{\mu_N}. \tag{52}$$

(For a planar curve, the case with $N = 2$ is the area $A[C]$ studied above). We could then consider a string field ansatz depending on all of these invariants, resulting in an infinite-dimensional (but conventional) PDE in this space. Using such techniques it should be possible to construct the string field propagator (which would compute off-shell string propagation amplitudes $\langle\psi(C_1)\psi^\dagger(C_2)\rangle$) and – more ambitiously – formulate perturbation theory.

We leave such considerations for the future and turn now to the spontaneously broken phase.

## 4 Spontaneously broken phase

We now consider the broken phase, i.e. we imagine that in the potential

$$V(\psi^\dagger\psi) = r\psi^\dagger\psi + \frac{u}{4}(\psi^\dagger\psi)^2, \tag{53}$$

we take $r < 0$, so that the minimum is at a nonzero value of $\psi$. Minimizing the potential $V$, we find a minimum with magnitude

$$v = \sqrt{\frac{-2r}{u}}. \tag{54}$$

The extremum at $\psi = 0$ is now unstable, and the action is minimized if we have a condensate in $\psi$, i.e. $\psi[C] = v$ for all $C$. Furthermore the mean-field dependence of $v$ on the parameter $r$ suggests that the transition is second-order, as usual for Landau-Ginzburg theory.

In applications to actual microscopic systems, we should note that the precise location of this minimum – and the order of the putative transition – is potentially sensitive to the topology-changing terms such as (27) that we have omitted in this study. We comment on this further in the conclusion.

Finally, we note that a string field with $\psi[C] = v$ for all loops is not strictly compatible with the boundary condition at infinitesimal loops (39); instead we expect that as the loop $C$ is made larger, $\psi[C]$ interpolates from some fixed microscopic value at small loops to the minimum of the potential set by $v$ in the infrared. As in the previous section, it is difficult to understand the form of this interpolation for general curves. To get some intuition, we numerically solve the nonlinear equations of motion arising from truncated action above (where we set $\lambda$ to zero for simplicity):

$$f''(a) - \frac{m\sqrt{\pi}}{\sqrt{a}} f'(a) - rf(a) - \frac{1}{2}uf(a)^3 = 0, \tag{55}$$

with $r < 0$ in Figure 4.

In particular, for large $C$, $\psi[C]$ is now completely independent of the curve $C$ and its magnitude is fixed at $v$

$$\psi[C] \approx v. \tag{56}$$

This long-distance behavior is the analogue of the factorized correlation function in the 0-form case (11), and clearly defines a different infrared phase than in the unbroken phase, where $\psi[C]$ vanished exponentially for large loops as the area law (37). This is the order parameter for a broken 1-form symmetry.

It is in fact a stronger statement than required. It may seem curious that $\psi[C]$ is completely independent of the curve $C$; usually when studying the order parameter for a spontaneously broken 1-form symmetry, one considers the symmetry to be broken – and obtains gapless Goldstone modes – if there is weak but still local dependence on $C$, e.g. a *perimeter* law for the charged line operator [5,9,25]. The complete independence on the curve $C$ is an extreme version of this, where the coefficient of the perimeter law has also vanished. It seems to us that this may be an artifact of our simplified model where we have ignored topology-changing terms such as (27).

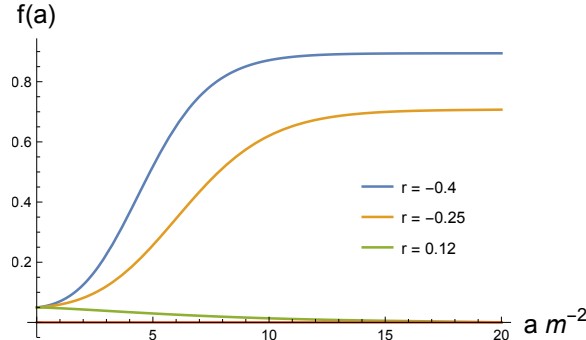

Figure 4: Some examples of different profiles of the string field as a function of the area in the condensed and uncondensed phase for different values of $rm^{-4}$

What are the low energy fluctuations around this condensed phase? We can obtain some intuition from the usual dynamics of Goldstone modes in 0-form symmetries. A Goldstone mode can be thought of as a space-time dependent symmetry transformation acting on the

order parameter. We are thus inspired to consider the following phase modulation of the string field:

$$\psi[C] = v \exp\left(i \int_C a(X)\right), \tag{57}$$

where $a_\mu(x)$ is a spacetime vector field that parameterizes the transformation. If $a$ is shifted by a closed form, it is precisely a 1-form symmetry transformation (13) acting on $\psi$; we thus expect $a$ to be the 1-form analogue of the Goldstone mode. The 1-form symmetry will ensure that the collective mode $a_\mu$ remains gapless, as we verify explicitly below.

Note that there is a redundancy in our description of the the string field ansatz in terms of $a$: if we shift $a \to a + d\Lambda$, then the string field is left unchanged. In conventional language, this would be called a gauge redundancy or by the oxymoron "gauge symmetry"; here it arises when trying to express the dynamical degree of freedom $\psi[C]$ in terms of a local spacetime field $a_\mu(x)$.

This is not the most general fluctuation around the vacuum. Fluctuations of the magnitude of the string field will be gapped, but phase fluctuations seem much less constrained. Even restricting to phase fluctuations that are local functions of the curve $C$, we should presumably consider at least the following more general form:

$$\psi[C] = v \exp\left(i \int_C ds \left(t(X) + a_\mu(X)\dot{X}^\mu + h_{\mu\nu}(X)\dot{X}^\mu \dot{X}^\nu + \cdots\right)\right), \tag{58}$$

where here $t(x)$, $a_\mu(x)$, $h_{\mu\nu}(x)$, and so on are *component fields* that make up the phase modulations of the original string field, i.e. we imagine that the path integral over the string field reduces to

$$[d\psi] = [dt\, da\, dh \cdots]. \tag{59}$$

It is possible to construct an effective action for each of these component fields in our formalism; in practice it is somewhat tedious and we will do so only for the field $t(x)$, demonstrating that it receives a mass, unlike $a_\mu$. There is no symmetry protecting any of the other modes, and we anticipate they will all be gapped.

## 4.1 World line path integrals

Below, for simplicity we will assume that we are working in a regime where the magnitude of the string field is constant, not worrying about the interpolation shown in Figure 4. To proceed, we will need to be more explicit about the integral over curves. We briefly review the (standard) logic: above we have been defining the integral to be weighted by an exponential of the proper length $L[C] = \int d\lambda \sqrt{\dot{X}}$ of the curve, i.e.

$$\int [dC] = \int [dX] \exp\left(-m \int d\lambda \sqrt{\dot{X}^2}\right). \tag{60}$$

As usual, the square root prevents us from performing the path integral in this representation; instead we assume that this is equivalent to the following representation, where we introduce an einbein $\epsilon(\lambda)$ along the curve:

$$\int [dC] = \int [dX d\epsilon] \exp(-S[X,\epsilon]), \qquad S[X,\epsilon] = \frac{1}{2}\int d\lambda \left(\epsilon^{-1}\dot{X}^2 + \epsilon m_0^2\right). \tag{61}$$

We have called the bare mass appearing in this action $m_0$, as it will soon receive quantum corrections. Classically, the equation of motion sets $\epsilon^2 = \dot{X}^2 m_0^{-2}$, and the two representations are

equivalent if we set $m_0 = m$. $\epsilon(\lambda)$ transforms under the gauge redundancy of reparametrizations of the curve. Gauge fixing the path integral, we find that the integral becomes [22, 27]:

$$\int [dC] = \int_0^\infty \frac{dL}{L} \int [dX] \exp\left(-\int_0^L ds \left(\frac{1}{2\epsilon} \dot{X}^2 + \frac{\epsilon}{2} m_0^2\right)\right), \tag{62}$$

where $\epsilon$ has been gauge-fixed to be a constant, and the path integral over it reduces to a 1d integral over the worldline modulus $L$, which can be interpreted as the length of the curve. We have renamed the parameter $\lambda \to s$ to make this explicit. Here we integrate over all periodic trajectories, i.e. $X^\mu(L) = X^\mu(0)$. Note that the value of the constant $\epsilon$ in the action appears arbitrary: by rescaling $s$ and then further redefining $L$ it can be removed from the integral entirely. This is true only if there are no UV divergences, as we will see shortly.

To warm up, let us explicitly compute the path integral (62) with no further dependence on $C$. The integral over the string modes $X$ is quadratic, and we find

$$\int [dC] = \int_0^\infty \frac{dL}{L} e^{-\frac{\epsilon m_0^2 L}{2}} \left[\det\left(\frac{1}{\epsilon} \frac{d^2}{ds^2}\right)\right]^{-\frac{d}{2}}. \tag{63}$$

This determinant is ill-defined, as the worldline action is invariant under translations, resulting in $d$ zero modes. Carefully separating out these zero modes into a separate integral $\int d^d x_0$ and computing the determinant, one finds the following answer for the $L$-dependence of the path integral [22]:

$$\int [dC] = N^{-1} \int d^d x_0 \int_0^\infty \frac{dL}{L} L^{-\frac{d}{2}} e^{-\frac{\epsilon m_0'^2 L}{2}}, \tag{64}$$

where $N$ is non-universal and $m_0'$ has been shifted relative to its bare value $m_0$ by a UV-divergent contribution. From now on we will call $m_0' \to m$, with $m$ the physical mass. Note that there is a further divergence arising from the $L \to 0$ limit; this is a UV divergence at small curves.

Let us now recall that our string field action as defined in (30) has a prefactor $\mathcal{N}$ that we have until now studiously avoided discussing. We finally confront it now. We pick $\mathcal{N}$ so that the path integral over the unit string field picks up *only* the spacetime volume, i.e.

$$\mathcal{N} \int [dC] \cdot 1 = \int d^d x_0, \tag{65}$$

(i.e. $\mathcal{N}^{-1} = N^{-1} \int_0^\infty \frac{dL}{L} L^{-\frac{d}{2}} e^{-\frac{\epsilon m_0'^2 L}{2}}$). In other words, we have chosen to absorb the ill-definedness of the measure of the integral over paths into the overall normalization of the action.

Finally, we will need the propagator along the worldline, i.e. the function $G(s, s')$ defined so that

$$\epsilon^{-1} \frac{d^2}{ds^2} G(s, s') = \delta(s - s'). \tag{66}$$

Here we note that this equation actually has no solution on the circle, which is compact. In the language of electrostatics, we cannot solve for the electric field of a point charge living in 1d if there is nowhere for the electric field lines to end. We follow the standard procedure [27] of adding a uniform neutralizing charge density of $L^{-1}$ to obtain the following modified equation:

$$\epsilon^{-1} \frac{d^2}{ds^2} G(s, s') = \delta(s - s') - \frac{1}{L}, \tag{67}$$

which can be directly solved in position space with periodic boundary conditions to give:

$$G(s,s') = \frac{\epsilon}{2}\left(|s-s'| - \frac{(s-s')^2}{L}\right). \tag{68}$$

Note that the coincidence limit $G(s \to s')$ is finite and equal to zero. This is an artifact of working in one dimension; in higher dimensions (as in a putative Landau-Ginzburg theory of $(p > 1)$-form symmetries) the propagator is of course singular in this limit.

## 4.2 Derivation of effective action for Goldstone mode

We begin by considering the simple ansatz (57) and seek to derive an effective action for the spacetime field $a_\mu$. The dynamics for this field will arise from the kinetic term in the string field action (30):

$$S[\psi]_{\text{kinetic}} = \mathcal{N}\int [dC]\left(\frac{1}{2L[C]}\oint_C ds \frac{\delta\psi^\dagger[C]}{\delta\sigma_{\mu\nu}(s)}\frac{\delta\psi[C]}{\delta\sigma^{\mu\nu}(s)}\right). \tag{69}$$

We consider the ansatz (57) and take the area derivatives using (21) to find

$$\frac{\delta}{\delta\sigma^{\mu\nu}(s)}\psi[C] = if_{\mu\nu}(X(s))\psi[C], \qquad f = da. \tag{70}$$

Inserting this into the string action, we see that we need to evaluate the following:

$$S[a] = \mathcal{N}\int [dC]\left(\frac{1}{2L[C]}\oint_C ds\, v^2 f_{\mu\nu}(X(s))f^{\mu\nu}(X(s))\right). \tag{71}$$

This is already essentially the desired structure. We are integrating the usual Maxwell kinetic term over a series of curves. As the two factors of $f_{\mu\nu}$ are evaluated at the same point, translational invariance requires that this be equivalent to an ordinary integral of $f^2$ over spacetime.

We now verify this by explicitly performing the integral over curves. For convenience, define $F(x) \equiv v^2 f_{\mu\nu}(x)f^{\mu\nu}(x)$ and its Fourier transform:

$$F(x) = \int \frac{d^d k}{(2\pi)^d} e^{ik\cdot x}\tilde{F}(k). \tag{72}$$

Inserting this decomposition into (71), we see that it takes the form

$$S[a] = \int \frac{d^d k}{(2\pi)^d}K(k)\tilde{F}(k), \tag{73}$$

where $K(k)$ is a form factor in momentum space that contains the information of the worldline integral, and is defined as:

$$K(k) \equiv \mathcal{N}\int \frac{dL}{L}[dX]\oint \frac{ds'}{2L}\exp\left(-\oint_0^L ds\left(\frac{1}{2\epsilon}\dot{X}^2 + \frac{\epsilon}{2}m_0^2\right) + ik\cdot X(s')\right). \tag{74}$$

We now perform the quadratic integral over $X$. Separating out the zero mode as before and using the parametrization of the determinant in (64) we find:

$$K(k) = \frac{\mathcal{N}}{N}\int \frac{dL}{L^{1+\frac{d}{2}}}\int d^d x_0 e^{+ik\cdot x_0}e^{-\frac{m^2 L^2}{2}}\oint \frac{ds'}{2L}\exp\left(-k_\mu k_\nu G^{\mu\nu}(s',s')\right), \tag{75}$$

where $G_{\mu\nu}(s,s') = \delta_{\mu\nu}G(s,s')$ with $G(s,s')$ defined in (68). We have worked this out in great detail for later purposes, but in reality it is rather trivial, as with coincident points $G(s',s') = 0$. Even if it had not been, the integral over the zero mode $x_0$ results in a delta function that sets $k \to 0$.

Using our judicious choice of normalization in (65), we then find that

$$K(k) = \frac{1}{2}(2\pi)^d \delta^{(d)}(k),\tag{76}$$

and from (73) we see that the effective action is simply

$$S[a] = \frac{v^2}{2}\int d^d x \left(f_{\mu\nu}(x)f^{\mu\nu}(x)\right),\tag{77}$$

i.e. a Maxwell kinetic term for the gauge field. This gapless mode is precisely the Goldstone mode for the spontaneous symmetry breaking of the higher-form symmetry. Here we see quite explicitly that it is possible to see this gapless mode emerge from a concrete modulation of the string field.

In general, a continuous 1-form symmetry means that there is a conserved 2-form current. What is the form of the 2-form current as a functional of the string field $\psi[C]$? In general, it takes the unwieldy form

$$J_{\mu\nu}(x) = \frac{\delta S}{\delta \mathcal{B}^{\mu\nu}(x)} = \int [dC]\int ds \left(\frac{\delta}{\delta\sigma_{\mu\nu}(s)} - i\mathcal{B}^{\mu\nu}\right)\psi^\star[C]i\delta(x - x(s))\psi[C] + h.c.$$

In the broken phase, this expression simplifies to $J_{\mu\nu}(x) = v^2(f_{\mu\nu}(x) + \mathcal{B}_{\mu\nu}(x))$, matching its expected form in Maxwell theory.

## 4.3 Other phase modulations are gapped

Here we explore the effect of modulating the phase in a different way, i.e. we study the effective action of the following scalar phase modulation:

$$\psi_t[C] = v\exp\left(i\int_C ds\, t(X)\right),\tag{78}$$

where $t(x)$ is a spacetime field. Note that $t(x)$ is not protected by any symmetry, and that the measure in the worldline integral – which we did not need to consider for the intrinsically reparameterization-invariant integral over $a$ in (57) – is given by the proper length along the curve.

We will follow the same steps as above. The required area derivative is worked out in (A.159):

$$\frac{\delta\psi_t[C]}{\delta\sigma^{\mu\alpha}(s)} = 2i\left(\frac{\dot{X}_{[\mu}\partial_{\alpha]}t(X(\lambda))}{\sqrt{\dot{X}^2}} - \frac{t(X(\lambda))\ddot{X}_{[\mu}\dot{X}_{\alpha]}}{(\dot{X}^2)^{\frac{3}{2}}}\right)\psi_t[C].\tag{79}$$

We first note that the equations of motion for the einbein set $\dot{X}^2 = m^2\epsilon^2$; thus within the path integral over $[dC]$ we may set these equal, up to contact terms that we will discard. We thus find that

$$\frac{\delta\psi_t[C]}{\delta\sigma^{\mu\alpha}(s)} = 2i\left(\frac{\dot{X}_{[\mu}\partial_{\alpha]}t(X(\lambda))}{m\epsilon} - \frac{t(X(\lambda))\ddot{X}_{[\mu}\dot{X}_{\alpha]}}{(m\epsilon)^3}\right)\psi_t[C].\tag{80}$$

Note that – unlike the gauge field case in (70) – there is a term here that depends on $t(x)$ with no derivatives. Indeed, a constant $t$ corresponds to a modulation by the proper length of the

curve, which has a non-vanishing area derivative as discussed around (A.150). We will now show that this 0-th derivative term in $t$ results in a mass for the field.

Finally, we note that there is no loss of generality in setting the $a$ field to zero in this section; any cross terms between the two will involve an odd number of derivatives of $\dot{X}$ and so will vanish.

Let us focus on the contribution of the (square of the) second term in (80) to the effective action. This takes the form

$$S_2[t] = 2\mathcal{N} \int [dC] \oint ds' \frac{1}{2L[C]} \left( v^2 t^2(X) \frac{\ddot{X}^\mu \ddot{X}_\mu}{(m\epsilon)^4} \right), \tag{81}$$

where we have used the fact that $\dot{X}^\mu \ddot{X}_\mu \sim \frac{d}{ds}(\dot{X}^2) = \frac{d}{ds}(m^2\epsilon^2) = 0$ in our gauge. As before, we define $T(x) = v^2 t^2(x)$ and denote its Fourier transform by $\tilde{T}(k)$; the action $S_2[t]$ is then $S_2[t] = \int \frac{d^d k}{(2\pi)^d} K_t(k) \tilde{T}(k)$, where (by steps precisely analogous to above), the form factor $K_t(k)$ is:

$$K_t(k) = \frac{2\mathcal{N}}{N(m\epsilon)^4} (2\pi)^d \delta^{(d)}(k) \int \frac{dL}{L^{1+\frac{d}{2}}} e^{-\frac{m^2 L^2}{2}} \oint \frac{ds'}{2L} \exp\left(-k_\mu k_\nu G^{\mu\nu}(s',s')\right) \langle \ddot{X}^\mu(s') \ddot{X}_\mu(s') \rangle. \tag{82}$$

The overall translational delta function allows us to set $k \to 0$ in the exponent. We thus only need to evaluate $\langle \ddot{X}^\mu(s') \ddot{X}_\mu(s') \rangle$. This is equal to:

$$\lim_{s \to s'} \frac{d^2}{ds^2} \langle X^\mu(s) \ddot{X}_\mu(s') \rangle = -d \lim_{s \to s'} \epsilon \frac{d^2}{ds^2} \delta(s-s'), \tag{83}$$

where we have used the definition of the propagator in (66), together with the fact that $\langle X^\mu(s) X^\nu(s') \rangle = -G(s,s')\eta^{\mu\nu}$.

We see that we need to evaluate two derivatives of a delta function at the origin. This is clearly singular and will depend on the regularization. For concreteness, let us regulate the delta function as a Gaussian:

$$\delta(x) = \lim_{\Delta \to 0} \frac{1}{\sqrt{2\pi}\Delta} e^{-\frac{x^2}{2\Delta^2}}, \tag{84}$$

where $\Delta$ is a UV cutoff on the worldline. We now find that $\delta''(0) \to -\frac{1}{\sqrt{2\pi}\Delta^3}$.

Assembling all of these pieces together and absorbing the integral over $L$ – which is the same – into $\mathcal{N}$ as before, we find

$$S_2[t] = \frac{1}{\sqrt{2\pi}} \frac{v^2 d}{m^4 (\epsilon\Delta)^3} \int d^d x \, t(x)^2. \tag{85}$$

As promised, this is a positive-definite mass term associated for the field $t(x)$. It is worth noting that a physical distance is formed by integrating the einbein $\epsilon$ along the worldline, and $\Delta$ is a quantity with units of worldline length; thus $\epsilon\Delta$ is a reparametrization-invariant distance, and we should identity this particular combination with a physical cutoff $\epsilon\Delta = \Lambda^{-2}$.

This calculation makes clear that the field $t(x)$ is gapped at the cutoff scale. To identify the physical correlation length associated with $t$ we should compute also its kinetic term. As by now the logic is clear, we relegate this computation to Appendix E. The final answer is:

$$S[t] = v^2 \int d^d x \left[ \left( \frac{1}{2} + \frac{1}{\sqrt{2\pi}} \frac{\Lambda^2}{m^2} \right) (\partial t)^2 + \frac{1}{\sqrt{2\pi}} \frac{\Lambda^6 d}{m^4} t(x)^2 \right]. \tag{86}$$

As there are more powers of the cutoff in the mass term than the kinetic term, it is clear that correlations of the $t(x)$ field are gapped at the cutoff scale. To write this in a more enlightening

way, it is helpful to consider the limit where $\Lambda \gg m$; in other words, the effective worldline tension is small enough to allow paths to contribute that are much larger than UV cutoff. In that case the action can be written in terms of the physical mass of the field $\mu^2$:

$$S[t] = \frac{v^2 \Lambda^2}{\sqrt{2\pi}m^2} \int d^d x \left( (\partial t)^2 + \mu^2 t^2 \right), \qquad \mu^2 = \frac{d\Lambda^4}{m^2}. \tag{87}$$

Heuristically, the area derivative of the scalar phase modulation involves many derivatives of the worldline; the integral over curves subjects the worldline to strong fluctuations at short distances, which are very sensitive to the value of $t(x)$, giving it a large mass. We anticipate that the effect will also gap out all the higher order fields in (58); indeed it is easy to see that the higher the spin, the more worldline derivatives are needed to soak up the indices.

## 4.4 Topological defects

Another important purpose of Landau-Ginzburg theory is in support of the topological classification of defects in the ordered phase (as reviewed in [28]). Here we initiate a study of the analogous story for mean string field theory.

On general grounds, we expect the existence of topological defects in the condensed phase, where the order parameter winds in some manner around a submanifold in which the symmetry is restored. For example in the 0-form case described by (10), we have a vortex in the condensate $\phi$. The vortex is a codimension-2 object that carries a topological charge measured by the winding number:

$$N = \oint dx^\mu \mathrm{Im}\, \phi^\dagger \partial_\mu \phi. \tag{88}$$

The phase field winds through $2\pi$ as we circle the vortex.

Here we discuss the corresponding structure for 1-form symmetries. For convenience of notation we restrict to $d = 4$. The low-energy action (77) is that of ordinary Maxwell electrodynamics in four dimensions. If we take the 1-form symmetry in question to enforce the conservation of electric flux, then the topological defects in the string field are magnetic monopoles, as we now explain.

For convenience we recall the Goldstone parametrization of the string field (57):

$$\psi[C] = v \exp\left( i \int_C a(X) \right). \tag{89}$$

Now consider $\mathbb{R}^4$ written in polar coordinates:

$$ds^2 = d\tau^2 + dr^2 + r^2 \left( d\theta^2 + \sin^2\theta \, d\phi^2 \right). \tag{90}$$

We now construct the string field $\psi_m[C]$ corresponding to having a monopole sitting at $r = 0$ and extending along $\tau$. For curves $C$ that do not come close to $r = 0$, this is given by (89), where the field $a_\mu$ takes on the usual monopole profile, which can be piecewise defined in the north and south hemispheres:

$$a^N = \frac{q_m}{2}(\cos\theta - 1)d\phi, \qquad a^S = \frac{q_m}{2}(\cos\theta + 1)d\phi. \tag{91}$$

Though these gauge fields appear different in space, if $q_m$ is appropriately chosen they define the same string field as a function of $C$. To be very concrete, consider evaluating $\psi_m[C]$ for these two gauge fields on a curve $C$ at fixed $(r, \theta, \tau)$ that goes through $2\pi$ in $\phi$. We have

$$\psi_{m,a^N}[C] = \psi_{m,a^S}[C] \exp(2\pi i q_m). \tag{92}$$

Thus, if $q_m \in \mathbb{Z}$, the string field is well defined. This is the usual Dirac quantization condition; here it arises naturally from the "UV completion" defined by the string field.

It is interesting to try and construct the analogue of the winding number. To do this, consider a family of curves $C_\vartheta$ that foliate the $S^2$, so that each point $X^\mu(\vartheta, \lambda)$ on the $S^2$ is identified by a choice of curve labeled by $\vartheta$ and a parameter along the curve labeled by $\lambda$. For example, one choice is to take each curve to be the orbit of $\phi$ at fixed polar angle $\theta = \vartheta$

$$\theta = \vartheta, \, r = 1, \, \phi = \lambda, \qquad \lambda \in [0, 2\pi). \tag{93}$$

The winding number can then be written as:

$$N = \oint_{S^2} d\vartheta d\lambda \frac{\partial X^\mu}{\partial \vartheta} \frac{\partial X^\nu}{\partial \lambda} \text{Im} \, \psi[C_\vartheta]^\dagger \frac{\delta}{\delta \sigma^{\mu\nu}(\lambda)} \psi[C_\vartheta]. \tag{94}$$

This is a higher-form analogue of (88).

Our discussion is not yet complete: as the Goldstone $a_\mu$ cannot be globally defined, the magnitude of the string field $\psi_m[C]$ must vanish at the core of the monopole. In the space of curves, this core is the set of $C$ that intersect the line $r = 0$. The 1-form symmetry is then restored at the core of the monopole. It would be very interesting to explicitly construct a $\psi_m[C]$ whose magnitude interpolates from zero to $v$.

The existence of monopoles in this EFT is consistent with the general lore that a compact $U(1)$ gauge theory (here obtained as the low-energy Goldstone description of a string field) will always come with dynamical magnetic monopoles [29]. Condensing such topological defects will destroy the condensate, returning us to the unbroken phase where line operators obey an area law; this provides a demonstration of the principle that condensation of magnetic monopoles is electric confinement.

Here is a more general perspective. If we choose a nice-enough submanifold $X$ of space in which the order parameter $\psi$ is known to be condensed (that is, $\psi[C] \neq 0$ for all loops $C$ restricted to $X$), it defines a map from the loop space[7] of $X$, $\Omega X$, to $U(1)$, the phase of $\psi$. This defines an element of

$$[\Omega X, S^1],$$

where the notation $[A, B]$ indicates the set of maps from $A$ to $B$ up to homotopy. If $\psi$ determines a nontrivial element of $[\Omega X, S^1]$, then there is a defect linked with $X$ that cannot be removed by deformations through continuous configurations on $X$. For example, if we take $X = S^2$ to be the sphere surrounding a point in $\mathbb{R}^3$, this object should classify the point defects we studied concretely above. To see this more explicitly[8], note that $S^1$ is an Eilenberg-MacLane space, $S^1 \simeq K(\mathbb{Z}, 1)$, which satisfies $[Y, K(A, n)] = H^n(Y, A)$ for any abelian group $A$, we have

$$[\Omega X, S^1] = [\Omega X, K(\mathbb{Z}, 1)] = H^1(\Omega X, \mathbb{Z}). \tag{95}$$

Now, in general $H^1(Y, Z)$ is the abelianization of $\pi_1(Y)$. But because $\pi_1(\Omega X) = \pi_2(X)$, $\pi_1$ of a loop space is already abelian. Therefore:

$$[\Omega X, S^1] = \pi_2(X). \tag{96}$$

In particular for $X = S^2$, there is a $\mathbb{Z}$ classification of defects, as expected.

An important choice of $X$ is $S^{q-1}$, which surrounds a codimension-$q$ defect in flat space. We conclude from (96) that mean string field theory admits only codimension-three defects.

---

[7]When studying $X$ which is not simply-connected, we must be more careful about choosing a base point.

[8]We are grateful to Iñaki García Etxebarria for advice here.

# 5 Discrete symmetries

All of our discussion up till now has discussed a continuous $U(1)$ symmetry. We now briefly consider the case of a discrete symmetry; as all higher form symmetries are Abelian [5] it is sufficient to consider the case of $\mathbb{Z}_N$. We warn the reader that the words below are somewhat tautological, as all of the universal information is encoded in the symmetry algebra already with little space for dynamics.

We can explicitly break the $U(1)$ symmetry considered above down to $\mathbb{Z}_N$ by adding a term to the potential of the form

$$S_N \equiv \int [dC]\, h\left((\psi[C])^N + (\psi[C]^\dagger)^N\right). \tag{97}$$

This new term has little effect in the unbroken phase; we still find an area law. We note that this is the case of interest for non-Abelian gauge theory with gauge group $SU(N)$, which has an unbroken $\mathbb{Z}_N$ 1-form symmetry in its confined phase.

There is an effect on the broken phase; in particular, the potential is now minimized at isolated minima where the phase is pinned to be at the $N$-th roots of unity,

$$\psi[C] = v \exp\left(\frac{2\pi i k}{N}\right), \qquad k \in 1\cdots N, \tag{98}$$

where $v$ is real. There is thus no more gapless Goldstone mode. There is however now a low-energy TQFT describing the broken symmetry phase. In particular, though $\psi[C]$ is pinned at a root of unity under small derformations of $C$, it is possible for topologically distinct $C$ to have different values for $\psi[C]$. As before, we parameterize this vacuum state by saying that

$$\psi[C] = v \exp\left(i \int_C a\right), \tag{99}$$

for a spacetime field $a$. However it is now required that $da = 0$, as otherwise $\psi[C]$ will depend continuously on $C$.

The universal information is encoded in the interplay between the $\mathbb{Z}_N$ charge operators and $\psi[C]$. We recall that a $\mathbb{Z}_N$ charge operator $U_k(\mathcal{M}_{d-2})$ is defined on a codimension-2 manifold. In the string field formalism we construct it by a modified boundary condition on the path integral:

$$\psi[C_1] = e^{\frac{2\pi i k}{N}} \psi[C_2], \tag{100}$$

where $C_1$ and $C_2$ are two nearby curves, except that $C_1$ links $\mathcal{M}_{d-2}$ (with positive orientation) and $C_2$ does not, as in Figure 5. (One can compare this to the 0-form co-dimension 1 charge operator, which also acts as a boundary condition on fields above and below it: $\phi(x^+) = \phi(x^-)\exp\left(\frac{2\pi i}{N}\right)$). By invariance of the action, the path integral is invariant under small deformations of $\mathcal{M}_{d-2}$, and thus $U_k(\mathcal{M}_{d-2})$ is a topological operator. By construction, it satisfies the following identity in all correlation functions:

$$U_k(S^{d-2})\psi[C] = e^{\frac{2\pi i k}{N}} \psi[C], \tag{101}$$

where $S^{d-2}$ is a small sphere wrapping $C$.

We now presume that the functional integral over the light degree of freedom $a$ in the absence of any insertions of $U_k(\mathcal{M}_{d-2})$ is

$$\int [d\psi]\exp(-S[\psi]) = \int [da\, dc]\exp\left(i \int \frac{N}{2\pi} c \wedge da\right), \tag{102}$$

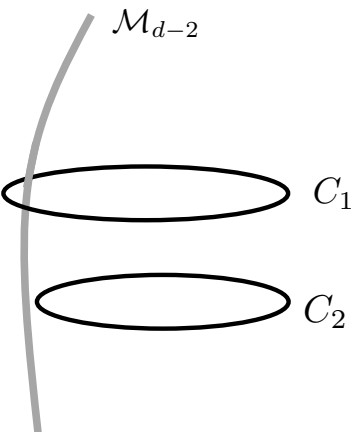

Figure 5: Example of linking surfaces in $d = 3$; here $C_1$ links the charge operator $\mathcal{M}_{d-2}$, but $C_2$ does not.

where $c$ is a new $d-2$ form Lagrange multiplier field introduced to force $da = 0$. The origin of $c$ may be understood as follows. In the broken phase, the perturbation (97) is the worldline representation of the path integral for a charge-$N$ particle coupled to the gauge field $a$, and the amplitude $e^{-S_N}$ exponentiates the disconnected diagrams. At large $h/m$, this model has a superfluid phase with order parameter $\phi$. The field $c$ is the dual, $dc = \star d\phi$, of the order parameter.

If we want to do the same path integral in the presence of the charge operator on $\mathcal{M}_{d-2}$, then we should write:

$$\int [d\psi] U_k(\mathcal{M}_{d-2}) e^{-S[\psi]} = \int [dadc] \exp\left( i \int \frac{N}{2\pi} c \wedge da \right) \exp\left( ik \int_{\mathcal{M}_{d-2}} c \right), \qquad (103)$$

which enforces the boundary condition (100). It is now interesting to consider the path integral in the presence of both a line operator insertion $\psi[C]$ and a charge operator, i.e. to compute:

$$\langle \cdots \psi[C] U_k[\mathcal{M}_{d-2}] \cdots \rangle. \qquad (104)$$

We see that this becomes

$$\int [dadc] \exp\left( i \int_c a + ik \int_{\mathcal{M}_{d-2}} c + \frac{iN}{2\pi} \int c \wedge da \right). \qquad (105)$$

We have arrived (as warned, in a somewhat tautological manner) at the low energy theory deriving the braiding statistics of line and charge operators. This is of course exactly the continuum description of $\mathbb{Z}_N$ gauge theory in its deconfined phase (see e.g. [30,31] for discussion in the high-energy physics literature).

In the case of a one-form symmetry with discrete group $\Gamma$, the classification of defects analogous to (95) is simpler to explain. An element of $[X, \Gamma]$ associates a group element of $\Gamma$ to each loop in $X$, with an ambiguity associated with the choice of base point, which acts by conjugation. But this is exactly the data of a flat $\Gamma$-bundle – a representation of $\pi_1(X)$ in $\Gamma$, up to conjugation. So defects linked with a manifold $X$ are classified (up to homotopy) by flat $\Gamma$ bundles on $X$ (up to homotopy).

# 6 On the consequences of emergent one-form symmetries

We now turn away from our specific framework to discuss some general principles bearing on the question of when we expect Mean String Field Theory to be useful. It is important to note that in many systems, we do not have *exact* higher-form symmetries, but rather approximate ones. For example, in our own universe 4d Maxwell EM has an apparent higher-form symmetry associated with the conservation of magnetic flux, and it has been argued that the photon is the massless Goldstone boson associated with the breaking of this symmetry. It is however quite probable that magnetic monopoles of high mass exist in our universe, thus explicitly breaking the 1-form symmetry at some high scale. When casting the photon as a Goldstone mode, it may now seem curious that this explicit breaking in the UV does not give the photon a small mass.

After all, this is not the case for familiar 0-form symmetries, where explicitly breaking a 0-form symmetry certainly gives the putative pseudo-Goldstone bosons a mass. A concrete example is the Gell-Mann-Oakes-Renner formula relating the pion masses in QCD to the explicit chiral-symmetry-breaking quark masses:

$$m_\pi^2 = \frac{(m_u + m_d)\langle \bar{q}q \rangle}{f_\pi^2}. \tag{106}$$

Note that even if we break such a 0-form symmetry only by irrelevant operators, we expect the (pseudo-)Goldstone bosons of emergent 0-form symmetries to have nonzero masses, albeit suppressed by a UV scale raised to a power determined by the dimension of the leading irrelevant operator that breaks the symmetry. Thus the Goldstone bosons of emergent 0-form symmetries are massive, yet the Goldstone bosons of emergent 1-form symmetries are not.

It appears, then, that there is an important and interesting distinction between the consequences of emergent one-form symmetries and those of emergent zero-form symmetries. We would now like to explore this phenomenon in more detail; as we will see, the string field formalism makes this particularly transparent, but we will present our arguments first in a more general language.

## 6.1 Review of 0-form deformations

Let us begin with a review of the situation for 0-form symmetries, in a somewhat lowbrow language. Consider a system with a 0-form $U(1)$ symmetry, which we deform at some scale $\Lambda$ by explicitly adding an operator of dimension $\Delta$ that breaks the symmetry to the action:

$$S_{\text{broken}} = S_0 + \Lambda^{d-\Delta} \int d^d x \left( \mathcal{O}(x) + \mathcal{O}^\dagger(x) \right). \tag{107}$$

Let us study this system in a box of size $L$. We may now ask when this deformation is important at long distances, i.e. at the scale $L$. The deformed partition function can be written as

$$Z_{\text{broken}} = \left\langle \exp\left( -\Lambda^{d-\Delta} \int d^d x \left( \mathcal{O}(x) + \mathcal{O}^\dagger(x) \right) \right) \right\rangle_0, \tag{108}$$

where the expectation value is taken in the undeformed system. Expanding in powers of $\Lambda$ the first potentially nonzero term is:

$$Z_{\text{broken}} = \langle 1 \rangle_0 + \left\langle \Lambda^{2(d-\Delta)} \int d^d x \, d^d y \left( \mathcal{O}(x)\mathcal{O}^\dagger(y) \right) \right\rangle_0 + \cdots. \tag{109}$$

Now let us ask when this integral is important in the limit $L \to \infty$. The answer to this question is determined by the two-point function in the undeformed theory, which is different for each possible realization of the symmetry:

1. **Spontaneously broken phase**: in this case, the two point function $\langle \mathcal{O}(x)\mathcal{O}^\dagger(y)\rangle_0 \sim v^2$ at large separations, factorizing into a separate integral over $x$ and $y$. We thus find that the integral scales at large $L$ as

$$\left\langle \Lambda^{2(d-\Delta)} \int d^d x\, d^d y\, \big(\mathcal{O}(x)\mathcal{O}^\dagger(y)\big)\right\rangle_0 \sim \Lambda^{2(d-\Delta)} L^{2d} v^2 . \tag{110}$$

This clearly *always* diverges as $L \to \infty$; thus we see that the deformation is *always* important in the infrared if we are deforming away from the broken phase. We cannot understand the precise effect in the IR from this crude reasoning, but a more careful analysis shows that it gaps out the Goldstone mode.

2. **Conformal point**: if we are sitting at a CFT, then we have $\langle \mathcal{O}^\dagger(x)\mathcal{O}(y)\rangle \sim |x-y|^{-2\Delta}$. Change variables to the sum and difference of $x$ and $y$; in the case the integral over the sum decouples and gives a factor of $L^d$. The integral over the difference then scales as

$$\Lambda^{2(d-\Delta)}\big(L^d a^{d-2\Delta} - L^{2(d-\Delta)}\big) , \tag{111}$$

with $a$ a short distance UV cutoff. Now the term in $a^{d-2\Delta}$ depends on $L$ in a local manner, and so can be canceled by a local counterterm. However the term in $L^{2(d-\Delta)}$ cannot be absorbed in that way and diverges whenever $\Delta < d$. We have reproduced the obvious fact that if we are sitting at the critical point, the deformation is important only if we deform by a relevant operator.

3. **Unbroken phase:** here we have $\langle \mathcal{O}^\dagger(x)\mathcal{O}(y)\rangle \sim e^{-m|x-y|}$. Repeating the same reasoning as above we find

$$L^d \Lambda^{2(d-\Delta)}\big(a^{d-2\Delta} - m^{2\Delta-d}\big) . \tag{112}$$

In this case, we see that the deformation happens at the scale of the mass gap $m$; in a strict sense we cannot discuss the extreme IR (as it is empty in any case), but physics at the scale of the original correlation length is expected to be altered.

The takeaway lesson here is that whether or not the deformation is relevant can be understood by a computation in the *undeformed* theory, i.e. the $L$-dependence of the integrated correlator $\left\langle \Lambda^{2(d-\Delta)} \int d^d x\, d^d y\, \big(\mathcal{O}(x)\mathcal{O}^\dagger(y)\big)\right\rangle_0$.

We have presented our arguments in a form that is independent of the description. However if we have access to a Landau-Ginzburg description of the system – as in (6) – these arguments can be made much more simply. Adding a term such as (107) corresponds to adding $\phi(x) + \phi^\dagger(x)$ to the Landau-Ginzburg action. In the condensed phase this combination becomes $v\cos(\theta)$, which clearly gaps out $\theta$. Similarly, we can explicitly study the effect on the phase transition and in the gapped phase and arrive at conclusions that agree with those given above.

## 6.2 1-form deformation

We now turn to 1-form symmetries. How does one break a 1-form symmetry? As in (6), we should add to the action an operator that is charged under the symmetry, i.e. a line operator $W(C)$ rather than a local operator $\mathcal{O}(x)$. In particular, the 1-form analogue of (108) is then

$$Z_{\text{broken}} = \left\langle \exp\left(-h \int [dX]\exp(-m_\Lambda L[C])W[C]\right)\right\rangle_0 , \tag{113}$$

where as before we replace $\int d^d x$ with a an integral over closed paths. The exponential of connected paths includes all possible disconnected loops.

This is a rather violent deformation; it corresponds to adding new degrees of freedom to the system, i.e. quanta that trace out 1d particle worldlines on which the 2d string worldsheets can end. A crucial and related difference from the 0-form case is that now one has to specify more data, corresponding to the dynamics of these new degrees of freedom. In particular, the integral over paths requires one to specify a line tension $m_\Lambda$, which corresponds to the scale at which the symmetry is broken. This deformation can be studied quite explicitly in the string field formalism; however, the arguments are rather more general, and we would first like to present them in a form that is independent of the description. We thus first proceed as above and study the dependence on the IR cutoff $L$ of the deformation (113). We then present a (rather simpler) discussion of the same physics in the string field formalism below.

### 6.2.1 Random walks on hypercubic lattice

For concreteness (and to obtain explicit control over the measure $\int [dX]$) let us take a model defined on a cubic lattice in $d$ dimensions. Denote the lattice distance of a given path $C$ to be $\ell[C]$. We take the sum over paths to be given by random walks on this lattice, weighted by a factor of $t_0^{\ell[C]}$:

$$\sum_C = \sum_{\text{paths}} t_0^{\ell[C]}, \tag{114}$$

where $t_0$ is the analogue of the (exponential of) $m_\Lambda$ in lattice units. Furthermore, let us assume that the relevant 1-form symmetry is spontaneously broken, so that $W[C]$ obeys a perimeter law, i.e. $\langle W[C]\rangle = t_W^{\ell[C]}$, where $t_W$ is a constant controlled by the dynamics of the undeformed system. We then expand (113) to get

$$Z_{\text{broken}}\left\langle 1 - h\sum_C W[C] + \cdots \right\rangle_0, \tag{115}$$

and we seek to understand the $L$-dependence of the second term.

This is a standard problem in statistical physics; see e.g. [32], whose treatment we follow. In particular, there is a transfer matrix approach to the sum over paths:

$$\left\langle \sum_C W[C] \right\rangle = L^d \frac{1}{2} \sum_{\ell=1}^\infty \left\langle 0 \left| \frac{t^\ell T^\ell}{\ell} \right| 0 \right\rangle, \tag{116}$$

where here $T$ is the transfer matrix, whose inner product satisfies:

$$\langle x|T|x'\rangle = \begin{cases} 1, & \text{if } x, x' \text{ are nearest neighbours}, \\ 0, & \text{otherwise}, \end{cases} \tag{117}$$

the hopping $t = t_0 t_W$ contains constributions from both the measure and the perimeter law, the state $|0\rangle$ is the origin of the lattice, and $\ell$ is the length of the loop. Diagonalizing the transfer matrix we find:

$$\left\langle \sum_C W[C] \right\rangle = -\frac{L^d}{2} \int \frac{d^d q}{(2\pi)^d} \log\left(1 - 2t\sum_{\alpha=1}^d \cos(q_\alpha)\right), \tag{118}$$

where $q_\alpha$ is a $d$-vector of momenta. There is a critical point at $t = t_c = \frac{1}{2d}$. Near the critical point we can define a correlation length as $\xi = \left(\frac{1-2td}{t_c}\right)^{\frac{1}{2}}$ and find that the integral at small $q$ goes like

$$\int d^d q \log(\xi^{-2} + q^2). \tag{119}$$

For $t < t_c$, this can be thought of as the normal 1-loop correction from a massive field, whose quanta trace out the random paths we have studied. Importantly, it is a harmless factor, independent of the IR cutoff $L$.

Thus, we have concluded that in the spontaneously broken phase, provided $t$ is small enough, the deformation by line operators will *not* have any effect in the IR, and in particular will not gap out any Goldstone modes that may be present.

This is quite different from the conclusion we arrived at for 0-form symmetries at (110), where we concluded that in the condensed phase *any* explicit breaking of the 0-form symmetry is very important in the IR. The crucial difference is the suppression of large paths in the partition function, as encoded in the bare worldline fugacity $t_0$; if this is small enough, then the 1-form deformation is irrelevant in the IR. In this sense there is an open set in parameter space that dos not affect the IR physics, and so it is "harder to break a 1-form symmetry".

On the other hand, the explicit lattice description lets us understand what happens when $t > t_c$; then the argument of the logarithm in (118) goes negative. In the present language the sum over the number of steps in (116) does not converge in this regime. In this toy model the random walks can pass through each other, and it seems that the sum is really infinite.[9] A more realistic worldline description would presumably involve self-avoiding walks. In any case there is an upper bound on the set of possible walks given by the size of the system; this is then an IR divergence, indicating that the explicit breaking of the 1-form symmetry is now relevant in the IR. In a field theoretical language we would say that the field (whose quanta follow the paths being summed) has condensed.

### 6.2.2 Explicitly breaking the symmetry with the string field

We now turn to a discussion of the same physics using the formalism of the string field. Recall that the string field action is

$$S_0[\psi; b] = \mathcal{N} \int [dX] e^{-mL[C]} \left( \frac{1}{2L[C]} \oint_C ds \frac{\delta \psi^\dagger[C]}{\delta \sigma_{\mu\nu}(s)} \frac{\delta \psi[C]}{\delta \sigma^{\mu\nu}(s)} + V(\psi^\dagger[C]\psi[C]) \right). \quad (120)$$

In particular, if we identify the line operator $W(C)$ with the string field $\psi[C]$, we see from (29) that the deformation (113) simply corresponds to adding the following term to the action:

$$S[\psi]_{\text{broken}} = S[\psi]_0 + \frac{h}{2} \int [dC] (\psi[C] + h.c.). \quad (121)$$

We may now rather easily understand the effect of this term. It is the special case of the perturbation we studied in §5 with $N = 1$. We discuss the 1-form analogues of the three scenarios outlined above in Section 6.1.

1. **Spontaneously broken phase:** We plug in the effective low-energy ansatz (57). We can explicitly do the integral over curves in $S[\psi]_0$, leading to the Maxwell kinetic term in (77). We find:

$$Z_{\text{broken}} = \int [da] \exp\left( \frac{v^2}{2} \int d^d x \left( f_{\mu\nu}(x) f^{\mu\nu}(x) \right) + h \int [dX] e^{-m_\Lambda L[C] + i \int_C a} \right). \quad (122)$$

   As expected, this is simply a Maxwell field coupled to (a worldline representation of) massive charged matter; provided $m_\Lambda > 0$, the matter decouples in the infrared and does not affect the IR dynamics or gap out the photon. This is simply a continuum version of the lattice arguments of the previous section, and similarly illustrates a key difference from the 0-form case. Note that the critical point at $t = t_c$ on the lattice corresponds to $m_\Lambda = 0$ here.

---

[9]This is the familiar fact that Bose condensation of *free* bosons at fixed chemical potential is singular.

2. **Critical point:** We do not currently have a precise understanding of the critical point at $r = 0$ in the Landau-Ginzburg action (120); nevertheless it is clear that at large $L$ the symmetry-breaking term in (121) is suppressed relative to the existing terms in the action, provided that $m_\Lambda > m$ (the bare worldline tension). Thus, if the deformation happens at a high scale, we again expect that the symmetry breaking is irrelevant at the critical point. This is again somewhat different from the situation for 0-form symmetries, where we would generically expect there to exist a relevant symmetry breaking deformation that would alter the dynamics at the critical point.

3. **Unbroken phase:** We can now examine the equations of motion following from the deformed action to find

$$-\frac{1}{2} e^{+mL[C]} \oint_C \frac{\delta}{\delta\sigma_{\mu\nu}(s)} \left( ds \frac{e^{-mL[C]}}{L[C]} \frac{\delta\psi[C]}{\delta\sigma^{\mu\nu}(s)} \right) + r\psi[C] = e^{-m'_\Lambda L[C]}, \qquad (123)$$

where $m_{\Lambda'} = m_\Lambda - m$. Here we will see an effect from the new term for sufficiently large curves. Recall that our WKB analysis earlier showed a solution to the string field of the form $\psi[C] \sim e^{-\sqrt{r}A[C]}$; thus the new term will be comparable to the existing size of $\psi$ when the minimal area $A$ satisfies:

$$\sqrt{r}A \sim m_{\Lambda'}\sqrt{A}, \qquad (124)$$

i.e. when the energy in the string tension is sufficient to pair produce particles. At that point we expect the area law to cross over to a perimeter law. Within this formalism it should be possible to explicitly derive the functional form of this crossover. Note that as $r \to 0$ the scale of this crossover is pushed out to arbitrarily large areas, again showing that the deformation is generically irrelevant at the critical point (see a recent discussion of this point on the lattice at [33]).

The above discussion is the beginning of an attempt at understanding the RG flow of line operators. We have discussed only gross qualitative features; there is clearly much more to be understood in this direction.

## 6.3 Finite temperature, 0- and 1-form symmetries, and topological order

A robust lesson from above is that it is generically *harder* to break a 1-form symmetry than a 0-form symmetry, providing an explanation for why emergent 1-form symmetries – such as (presumably) magnetic flux conservation in our own universe – still usefully constrain infrared physics.

It is interesting to consider compactifying a system with a 1-form symmetry on an $S^1$ with length $R$. Denoting the coordinate on the $S^1$ by $\tau$, the underlying 1-form symmetry with current $J^{\mu\nu}$ then decomposes into a 0-form symmetry with $J^{i\tau}$ and a 1-form symmetry with current $J^{ij}$. This has implications for finite-temperature physics. Let us consider 4d Maxwell electromagnetism with dynamical electric charges with mass $m_e$ as an example[10]. On $\mathbb{R}^4$, this system has an exact 1-form symmetry for magnetic flux conservation, and an emergent 1-form symmetry for electric flux conservation, by the arguments above.

However on $S^1 \times \mathbb{R}^3$, if we integrate out the electric matter in the limit $m_e R \to \infty$, the low-energy action for the gauge field takes the form

$$S_{3d}[a_i, a_\tau] = \int d^3x \left( \frac{1}{4} f_{ij} f^{ij} + \frac{1}{2} (\partial_i a_\tau)^2 + c_e e^{-m_e R} \cos(a_\tau R) + \cdots \right), \qquad (125)$$

---

[10]This symmetry-breaking pattern has recently attracted attention in connection with formulations of relativistic magnetohydrodynamics in terms of higher-form symmetry [34–36]; see [37] for a discussion in language similar to that used here.

where $f_{ij} = \partial_i a_j - \partial_j a_i$. Note the term arising from the electric worldlines wrapping the thermal cycle; these give a thermal mass $e^{-m_e R}$ to the time component of the gauge field, explicitly breaking the 0-form shift symmetry. (In more conventional language, this is Debye screening of the vector potential). By reducing part of the 1-form symmetry to a 0-form symmetry, we have made it easier to break. Note that as we decompactify the thermal cycle the mass vanishes exponentially. The 1-form symmetry associated with the conservation of electric flux $\partial_i f^{ij} = 0$ however remains emergent at long distances.

We could also consider the situation with both dynamical electric and magnetic charges with mass $m_e$ and $m_m$ respectively. We should now dualize the 3d vector potential above to a 3d compact scalar $\sigma$ via $da = \star_3 d\sigma$; summing over both magnetic and electric charges wrapping the thermal cycle, the low-energy action becomes

$$S_{3d}[\sigma, a_\tau] = \int d^3x \left( \frac{1}{2}(\partial_i \sigma)^2 + \frac{1}{2}(\partial_i a_\tau)^2 + c_e e^{-m_e R} \cos(a_\tau R) + c_m e^{-m_m R} \cos \sigma + \cdots \right). \quad (126)$$

We see that now all the components of the 3d photon are gapped. The 4d magnetic monopoles that wrap the thermal cycle appear in the dimensionally reduced theory as 3d magnetic monopole instantons, which confine the 3d gauge field via the usual Polyakov mechanism [38, 39] explicitly breaking the magnetic 0-form shift symmetry. (Physically this corresponds to a thermal gas of magnetic monopoles and electric charges which disorder all long-range correlations).

We see that finite temperature results in the destruction of long-range correlations by providing a 0-form symmetry that can be explicitly broken. This brings out an interesting parallel with topological order, *i.e.* the case where the 1-form symmetry is discrete. Recall that known forms of topological order in $d \leq 3+1$ have the property that at any finite temperature they are smoothly connected to $T = \infty$, *i.e.* to a trivial product state [40–42]. The argument above is perfectly consistent with this fact: if the 1-form symmetry is emergent, then as soon as $T > 0$, a mass *is* generated for (all the components of) the photon, and the state is smoothly connected to $T = \infty$.

This further raises the interesting point that we do know an example of a topologically ordered phase which is stable at $T > 0$, namely the 2-form toric code in 4+1d [40, 41, 43]. If we consider the U(1) version of this theory, this suggests that the masslessness of the 2-form gauge field should survive explicit short-distance breaking of the U(1) 2-form symmetry even at finite temperature $0 < T < T_c$. Indeed, when we put a theory with a 2-form symmetry on a circle, it still has a 1-form symmetry, which does not break under deformation by line operators, by the arguments above. We thus conclude that the deconfined phase of 4d 2-form gauge theory survives finite temperature. It would be very interesting to revisit from this point of view the phase transition at finite $T_c$ where the topological order is finally destroyed by proliferation of strings.

# 7 What is it good for?

We briefly summarize our results here: motivated by considerations of effective field theory and higher-form symmetry, we introduced a mean string field theory which plays the same role for 1-form symmetries as conventional mean field theory does for 0-form symmetries. In this final section we briefly sketch some potential applications, as well as highlighting what we see as the caveats in our analysis.

**Lack of universality:** In this work we studied mean string field theory with a simple local potential energy for the string field, which took the form

$$\int [dC]\left(r\psi^\dagger\psi + \frac{u}{4}(\psi^\dagger\psi)^4 + \cdots\right). \tag{127}$$

We studied the theory as the sign of $r$ was altered, as normal for Landau-Ginzburg theory of 0-form symmetries. In particular, if $r < 0$, we found a condensed vacuum by balancing the self-interaction $u$ against the tension term $r$.

However, a new ingredient in the Landau-Ginzburg theory of 1-form symmetries is the possibility of terms such as (27):

$$S_{\text{tc}} = \lambda \int [dC_1][dC_2][dC_3]\delta[C_1 - (C_2 + C_3)]\psi^\dagger[C_1]\psi[C_2]\psi[C_3] + \text{h.c.} + \cdots. \tag{128}$$

Such terms appear to be allowed on symmetry grounds. We did not study them in the bulk of the paper, primarily because they are rather complicated and we have not yet been able to treat the delta function in loop space systematically. It seems possible that a proper analysis will result in new condensed vacua from balancing the tension term $r$ against this topology-changing term.

An important observation here is that these new terms are cubic in the string-field $\psi[C]$; the addition of a cubic term to a Landau-Ginzburg potential generically has the effect of changing the order of the transition from second order to first order. Interestingly, this state of affairs appears broadly consistent with numerical studies on the lattice, which suggest that most confinement/deconfinement transitions in lattice gauge theory in sufficiently high dimension are first-order (see e.g. [44–50]). The situation is somewhat complicated by the fact that there nevertheless exist examples of continuous transitions in *low* dimension, discussed below. We do not yet have a unified picture of the order of the transition in different dimensions, but we believe this could be obtained through an understanding of the effects of terms such as (128).

A further shortcoming of our analysis is the form of the couplings themselves. As discussed around (25) and further elaborated on in Appendix B, all of the couplings that are present in the action could in principle be functions of the invariant length of the curve $L[C]$ (or area derivatives of $L[C]$). We have chosen a particular set of couplings that have no such functional dependence. This is a perfectly fine model to study, but from the principles of effective theory it appears finely-tuned. E.g. in general the coefficient of the mass term $r$ could instead take the form:

$$r(L[C]) = r_0 + \frac{r_1}{\Lambda L[C]} + \frac{r_2}{(\Lambda L[C])^2} + \cdots, \tag{129}$$

where we have chosen to expand it about $L \to \infty$, and where $\Lambda$ is some UV scale. Though such terms appear irrelevant, we have not tried to understand their significance, and they could be important at the phase transition. We imagine that they would be generated if we take fluctuations into consideration.

On the other hand, it is also possible that some symmetry principle – more refined than that discussed in Appendix B – constrains this functional dependence. A candidate for such a symmetry principle would be the (ordinary) spatial locality of the underlying microscopic Hamiltonian. We do not understand at the moment how conventional locality affects dynamics in loop space.

To explore all of these issues of universality, it would be very interesting to directly derive the mean string field theory from a more conventional lattice description. We have performed some preliminary investigation of variational wavefunctions parametrized by a string field for the deformed $\mathbb{Z}_2$ toric code in 2d [51–53] finding a variational energy that is structurally

similar to (30), with no extra dependence on $L[C]$, but with topology-changing terms similar to (128). We hope to report on this in the future [26].

**Upper critical dimension:** We now return to the theory described only by the local potential (127), which does have a continuous transition at $r = 0$. Let us determine the upper critical dimension $D$ of our theory. Recall from our definition of the normalization of the action (65) $\mathcal{N} \int [dC] = \int d^d x_0$ that the dimension of the measure of integration arises from the integral over the string zero modes. From the kinetic term we see that the (mass) dimension of the string field is then

$$[\psi] = \frac{d-4}{2}. \tag{130}$$

(Compare with $[\phi] = \frac{d-2}{2}$ for field theory; the difference arises from the dimension of the area derivative, which is 2). We can now attempt to use dimensional analysis to suppress higher derivative terms as usual in conventional field theory. There are two types of interactions, local and non-local, as discussed around (27). Studying first the local interaction, we see that the dimension of the coupling $u$ in the leading $|\psi|^4$ interaction is $8 - d$. Thus if we were to consider only the local potential (127), the upper critical dimension of our mean string field theory would be $D = 8$.

We expect this conclusion to be generically altered by the presence of the topology-changing interaction (128), and thus the broader significance of this result is not clear. Nevertheless, let us compare with results in the statistical mechanics of random surfaces, i.e. the first-quantized picture of the same problem. It was argued by Parisi [54] somewhat heuristically that the Hausdorff dimension of a random surface is 4, and thus that two random surfaces will generically intersect – rendering interactions between them relevant – in $d < 8$. (Note the same argument applied to random *walks* correctly predicts the upper critical dimension of scalar field theory to be 4 [55]). Interestingly, this precisely agrees with our result above. An explicit analysis in a particular ensemble by Gross [56] argued instead that the Hausdorff dimension (and thus the upper critical dimension) is infinite. Within our framework, we have been unable to think of an argument resulting in an infinite upper critical dimension. It seems possible that different ensembles over random surfaces result in different sets of constraints on possible string interactions. It would be extremely interesting to understand this further.

**Critical exponents:** We now discuss the critical exponents of our theory. We introduce an exponent $\mu$ for the vanishing of the string tension at a continuous critical point tension $\sim (T - T_c)^\mu$. Let us first review expectations from conventional field theory, organized around particle-like excitations. In that case we expect a correlation length $\xi$ to diverge at the transition with a critical exponent that is usually called $\nu$, i.e. $\xi \sim (T - T_c)^{-\nu}$. On dimensional grounds, we might expect to find $\mu = 2\nu$; in the context of interfaces at critical points this relationship is called Widom's scaling law [57]. Thus for ordinary mean-field theory of particles we expect:

$$\nu_{\text{MFT}} = \frac{1}{2}, \qquad \mu_{\text{MFT}} = 1. \tag{131}$$

Note this is very different from the prediction for mean *string* field theory; there the particle correlation length and associated critical exponent $\nu$ is not a fundamental object, but the string tension is. As shown explicitly in (38), we thus find instead that

$$\mu_{\text{MSFT}} = \frac{1}{2}. \tag{132}$$

Thus it seems that for a given phase transition one can ask whether it has a "free-particle-like" or a "free-string-like" character.

Now let us compare to explicit lattice examples. One such is the continuous confinement/deconfinement transition of $\mathbb{Z}_2$ gauge theory in 3d, which is in the same universality

class as the usual 3d Ising spin model [2]. This has been extensively studied numerically; see [58] for a review of the data on the interface scaling exponent $\mu$. One finds that

$$\mu_{\text{Ising}} \approx 1.26 \,, \tag{133}$$

consistent with Widom's law applied to the usual 3d Ising exponent $\nu_{\text{Ising}} \approx 0.6299$. Thus – perhaps unsurprisingly – the 3d $\mathbb{Z}_2$ model is more closely approximated by a free particle description than by a free string description. (Of course it is strongly coupled even in its particle description). This is entirely consistent with previous attempts at describing the 3d Ising model as a string theory, where it appears that a large string coupling $g_s$ is an unavoidable feature of the description [23, 59].

We are unaware of many other examples of continuous transitions for the breaking of a 1-form symmetry in higher dimension. A natural field-theoretical model with the appropriate phases is the Higgsing (or – equivalently after EM duality – confinement) of a $U(1)$ gauge theory in $d = 4$; unfortunately for the potential application of our theory, the phases of spontaneously broken and unbroken symmetry are separated by a transition that is weakly first order [60] (see e.g. [48] for a demonstration on the lattice). One may view our continuum description as motivating a search for other continuous transitions on the lattice (see a recent study in $d = 5$ in [50]), keeping in mind that the presence of terms such as (128) may indicate that accessing such a transition may require us to tune more than one coupling to zero.

**Higher dimensional critical points:** Another motivation to search for such a continuous critical point on the lattice arises from the fact that Lagrangian field theory is generically weakly coupled in the infrared in $d > 4$; thus any non-trivial critical point will necessarily not have a Lagrangian description in terms of field theory. In particular, any UV fixed point of the renormalization group above four dimensions must be something other than a gauge theory. There are many quantum field theories with no known Lagrangian description, most of which arise from string theory constructions, such as worldvolume theories of M5-branes (for reviews, see e.g. [61–63]), though recently a large class of such theories was proposed with plausible condensed matter realizations [64].

This situation motivates the study of frameworks that extend our understanding of how to formulate a field theory. Given an upper critical dimension $D$ for our theory, it seems that in in principle one could perform an $D - \epsilon$ expansion to describe entire families of new critical points, which might even conceivably include some of the theories discussed above.

It is interesting to place this in the context of usual critical phenomena. If one wants to study (e.g.) the breaking of an $O(N)$ 0-form symmetry, there are (at least) two ways to do it; one can build an expansion in $d = 2 + \epsilon$, expanding around the lower critical dimension by realizing the symmetry non-linearly in terms of a non-linear sigma model, which is asymptotically free in the UV at $d = 2$. Alternatively, one can build an expansion in $d = 4 - \epsilon$, realizing the symmetry linearly in terms of a linear sigma model. It appears that similar words can be said about (e.g.) a $\mathbb{Z}_N$ 1-form symmetry; one can realize the symmetry non-linearly in terms of an $SU(N)$ gauge theory, which is asymptotically free in the UV in $d = 4$, and build an expansion about the lower critical dimension in $d = 4 + \epsilon$ [65, 66]. Alternatively, one can realize the symmetry linearly in terms of the string field developed here, and perform an expansion about the upper critical dimension $d = D - \epsilon$. The existence of a gauge theory analogue of the 0-form $4 - \epsilon$ expansion was postulated long ago by [54], but to the best of our knowledge this mean string field theory is the first proposal as to its identity, made possible by a careful identification of the global symmetries.

**A little philosophy:** We conclude by noting speculatively that the very existence of gauge theory (as conventionally formulated) is somewhat peculiar; after all, why should a description in terms of (often frustratingly) redundant variables be physically useful? One way to understand this is that gauge theory provides a local framework to describe the dynamics of

extended objects such as flux tubes or magnetic field lines, and that it is the integrity of these extended objects that is fundamental, not the gauge fields. This idea lies behind the loop formulation of non-Abelian gauge theory [10–12]. The line of reasoning explored in our work is an attempt at organizing the dynamics *entirely* around this viewpoint, eschewing the microscopic gauge fields entirely. Clearly there is still much to be understood. It remains to be seen if such ideas will help us liberate our understanding of gauge theories from the tyranny of gauge redundancy.

## Acknowledgements

It is a pleasure to acknowledge helpful discussions on related issues with Andreas Braun, Iñaki García Etxebarria, Tarun Grover, Diego Hofman, Napat Poovuttikul, Tin Sulejmanpasic and David Tong. We thank Zhengdi Sun for helpful comments on the first draft. NI is supported in part by the STFC through grant ST/P000371/1. This work was supported in part by funds provided by the U.S. Department of Energy (D.O.E.) under cooperative research agreement DE-SC0009919, and by the Simons Collaboration on Ultra-Quantum Matter, which is a grant from the Simons Foundation (JM, 651440).

## A    Area derivatives

Here for completeness we explicitly compute a few area derivatives. All but the derivative of the scalar field coupling can be found in [11, 20].

1. **Minimal area** Let $A[C]$ denote the area of the minimal surface $S$ that "fills in" the contactible curve $C$, i.e. $\partial S = C$. Its area derivative can be computed to be:

$$\frac{\delta A}{\delta \sigma^{\mu\nu}(\lambda)} = n_\mu t_\nu - n_\nu t_\mu, \tag{A.134}$$

where $n$ is the *outward* pointing normal and $t_\nu$ is the tangent vector to the curve (normalized to have norm 1). This is somewhat clear geometrically, but we also present an analytic derivation following [11]. Consider writing the minimal area in terms of a double integral over the minimal surface as

$$A[C] = \lim_{\Lambda \to \infty} \left( \frac{1}{2}\Lambda^2 \int_S d\sigma^{\mu\nu}(x) \int_S d\sigma_{\mu\nu}(y) G(\Lambda^2(x-y)^2) \right). \tag{A.135}$$

Here $G$ is a regulator function that satisfies the following normalization condition when integrated over two dimensions:

$$\int d^2 y\, G(y^2) = 1. \tag{A.136}$$

Now we further note that we can write

$$d\sigma^{\mu\nu}(x) = 2n^{[\mu} t^{\nu]} d^2 x, \tag{A.137}$$

where $d^2 x$ is the usual proper area measure on the surface $S$ and $n^\mu$ and $t^\nu$ form a basis for two normalized vectors that span its tangent plane at the point $x$. From here it is straightforward to verify that in (A.135) we have

$$A[C] = \frac{1}{2}\int_S d\sigma^{\mu\nu}(x) 2(n^{[\mu} t^{\nu]}(x)) = \int_S d^2 x, \tag{A.138}$$

as desired. As $\Lambda \to \infty$ the regulator becomes a delta function and we receive a contribution only when the two points coincide, which measures the proper area along the surface.

Now consider varying $C$ by adjoining a small extra curve $\delta C$ at a point $x_C$ on $C$. From the variation of (A.135) we then have

$$\delta A[C] = \lim_{\Lambda \to \infty} \left( \Lambda^2 \delta\sigma^{\mu\nu}(x_C) \int_S d\sigma_{\mu\nu}(y) G(\Lambda^2 (x-y)^2) \right) = \delta\sigma^{\mu\nu}(x_C)(n_\mu t_\nu - n_\nu t_\mu)\big|_{x_C}, \tag{A.139}$$

which leads to (A.134) as claimed. Note in particular that the normal and tangent vectors to $C$ provide a basis for the tangent space of the minimal surface $S$ at $C$.

2. **Proper length**

A less straightforward application of the technology is to the proper length functional $L[C]$:

$$L[C] \equiv \int d\lambda \sqrt{\dot{X}^2}. \tag{A.140}$$

Again for completeness we record the computation of [10]). Here we consider a regulator function $F$ such that

$$\int_{-\infty}^{+\infty} d\xi F(\xi^2) = 1, \tag{A.141}$$

and we then write the proper length as

$$L[C] = \lim_{\Lambda \to \infty} \oint_C dX_\mu \oint_C dY^\mu \Lambda F(\Lambda^2 (X-Y)^2). \tag{A.142}$$

Here both integrals are done over the same curve, and the regulator is a function of the distance between the two points.

Now we consider

$$\Delta L \equiv L[C \cup \delta C] - L[C] = 2\Lambda \oint_{\delta C} dX_\mu \int_C dY^\mu F(\Lambda^2 (X-Y)^2). \tag{A.143}$$

Denote the insertion point of $\delta C$ by $X_0$. We now want to extract the part of this which is proportional to the infinitesimal area element. To do this we Taylor expand the regulator in powers of $(X - X_0)$.

$$\Delta L = 2\Lambda \oint_{\delta C} dX_\mu \oint_C dY^\mu \left( F(\Lambda^2 (X_0 - Y)^2) + \partial_\alpha F(\Lambda^2 (X_0 - Y)^2)(X - X_0)^\alpha + \cdots \right). \tag{A.144}$$

The leading term vanishes when integrated around the small loop, and the next term generates the area element via (16):

$$\Delta L = 4\Lambda \sigma^{\mu\alpha}(\delta C) \oint dY_{[\mu} \partial_{\alpha]} F(\Lambda^2 (X_0 - Y)^2). \tag{A.145}$$

We now need to compute this integral. It is helpful to parametrize the curve with a parameter $s$ that measures proper length, and so that $X(s_0) = X_0$. Then we have

$$Y^\mu = X_0^\mu + t^\mu(s-s_0) + \frac{1}{2}\dot{t}^\mu(s-s_0)^2 + \cdots, \qquad dY^\mu = ds\,(t^\mu + \dot{t}^\mu(s-s_0) + \cdots), \tag{A.146}$$

where $t^\mu$ is the tangent vector to the curve (and thus the expansion above is simply the expansion in powers of $(s - s_0)$). We also have

$$\partial_\alpha F(\Lambda^2(X_0 - Y)^2) = 2\Lambda^2(X_0 - Y)_\alpha F'(\Lambda^2(s - s_0)^2) = 2\Lambda^2\left(t_\alpha(s - s_0) + \frac{1}{2}\dot{t}_\alpha(s - s_0)^2\right)F'. \tag{A.147}$$

Putting these together and performing the antisymmetrization, we find

$$\Delta L = 4\Lambda^3\sigma^{\mu\alpha}\dot{t}_{[\mu}t_{\alpha]}\int ds(s - s_0)^2 F'(\Lambda^2(s - s_0)^2), \tag{A.148}$$

which we rewrite as

$$\Delta L = 2\Lambda\sigma^{\mu\alpha}\dot{t}_{[\mu}t_{\alpha]}\int ds(s - s_0)\frac{d}{ds}F(\Lambda^2(s - s_0)^2) = -2\sigma^{\mu\alpha}\dot{t}_{[\mu}t_{\alpha]}, \tag{A.149}$$

where in the last line we have integrated by parts.[11]

We conclude that the area derivative is

$$\frac{\delta L}{\delta\sigma^{\mu\alpha}(\lambda)} = -2\dot{t}_{[\mu}t_{\alpha]} = t_\alpha\dot{t}_\mu - t_\mu\dot{t}_\alpha, \tag{A.150}$$

where $t = \dot{X}^\mu$ is the tangent vector and where all derivatives are taken with respect to proper length. It is straightforward to write this in a reparam-invariant manner by adding appropriate facotors of $\dot{X}^2$ if needed.

Finally, let us note that as $L$ is written in terms of a general parameter $\lambda$ as $L = \oint d\lambda'\sqrt{\dot{X}^2(\lambda')}$, locality on the worldline and the equation above appear to imply:

$$\frac{\delta}{\delta\sigma^{\mu\alpha}(\lambda)}\sqrt{\dot{X}^2(\lambda')} = -\frac{2}{\sqrt{\dot{X}^2}}\dot{t}_{[\mu}t_{\alpha]}\delta(\lambda - \lambda'). \tag{A.151}$$

Integrating both sides over $\lambda'$, we obtain (A.150). This is a somewhat formal expression that we will use later.

3. **Scalar coupling**

We will be interested in the following slight generalization of the proper length:

$$\phi_T[C] = \oint_C d\lambda\sqrt{\dot{X}^2}T(X), \tag{A.152}$$

where $T(X)$ is now a space-dependent field. (Note that to avoid confusion with the tangent vector, we have called the scalar field $t(x)$ in the bulk of the text $T(X)$ in this Appendix.) To compute its area derivative, we regularize as above:

$$\phi_T[C] = \lim_{\Lambda\to\infty}\oint_C dX_\mu\oint_C dY^\mu T(X)\Lambda F(\Lambda^2(X - Y)^2). \tag{A.153}$$

We then follow the same steps as above. We have

$$\Delta\phi_T[C] = \oint_{\delta C} dX_\mu\oint_C dY^\mu (T(X) + T(Y))\Lambda F(\Lambda^2(X - Y)^2). \tag{A.154}$$

---

[11]The final sign appears to disagree with [10]; this is not important for our purposes, but we believe the sign recorded in this work is correct.

Now we expand the right hand side about $X_0$ as before to find

$$\Delta\phi_T[C] = 2\Lambda\sigma^{\mu\alpha}(\delta C)\oint dY_\mu\Big(\partial_\alpha F(\Lambda^2(X_0-Y)^2)(T(Y)+T(X_0))$$
$$+\partial_\alpha T(X_0)F(\Lambda^2(X_0-Y)^2)\Big). \tag{A.155}$$

The term without any derivatives on $t$ will give the same expression as above. We thus focus on the term that has a derivative of $t$:

$$\Delta\phi_T[C]' = 2\Lambda\sigma^{\mu\alpha}(\delta C)\oint dY_{[\mu}\partial_{\alpha]}T(X_0)F(\Lambda^2(X_0-Y)^2). \tag{A.156}$$

We can now expand $dY_\mu$ using (A.146); the only term that survives in the $\Lambda\to\infty$ limit is

$$\Delta\phi_T[C]' = 2\sigma^{\mu\alpha}(\delta C)t_{[\mu}\partial_{\alpha]}T. \tag{A.157}$$

Thus we can conclude that the functional derivative of the whole expression is

$$\frac{\delta\phi_T[C]}{\delta\sigma^{\mu\alpha}(\lambda)} = 2t_{[\mu}\partial_{\alpha]}T(X(\lambda))-2T(X(\lambda))\dot{t}_{[\mu}t_{\alpha]}. \tag{A.158}$$

Finally, this was in a parametrization where $\dot{X}^2=1$; if we want to write this in arbitrary parametrization it becomes

$$\frac{\delta\phi_T[C]}{\delta\sigma^{\mu\alpha}(\lambda)} = \frac{2\dot{X}_{[\mu}\partial_{\alpha]}T(X(\lambda))}{\sqrt{\dot{X}^2}}-\frac{2T(X(\lambda))\ddot{X}_{[\mu}\dot{X}_{\alpha]}}{(\dot{X}^2)^{\frac{3}{2}}}. \tag{A.159}$$

# B Translational invariance in the space of curves

Here we discuss a notion of "translational" invariance in the space of curves. Let us first express in an overly formal but instructive manner the constraints that translational invariance places on the action of a usual local quantum field theory. Consider a field theory with a local field $\phi(x)$. Under a translation $x^\mu\to x^\mu+\xi^\mu$ with $\xi^\mu$ constant, the transformation of $\phi$ is

$$\phi(x)\to\phi(x)+\xi^\mu\partial_\mu\phi(x). \tag{B.160}$$

We now demand that the action $S[\phi]$ should be invariant under this transformation. Let us consider for example an action of the following form:

$$S[\phi] = \int d^dx\,h(x)\phi(x), \tag{B.161}$$

where we imagine a putative space-dependent coupling $h(x)$. Under the transformation (B.160), we find that

$$\delta_\xi S[\phi] = \int d^dx\,h(x)\xi^\mu\partial_\mu\phi(x) = \int d^dx\left[\partial_\mu(\xi^\mu h(x)\phi(x))-\xi^\mu(\partial_\mu h(x))\phi(x)\right]. \tag{B.162}$$

The first term vanishes as it is a total derivative. The second term vanishes for arbitrary $\phi(x),\xi$ only if $\partial_\mu h(x)=0$. We thus conclude that all couplings in the action must be independent of space. Note that this argument required $\xi$ to be constant. It is possible to make a stronger statement involving more general $\xi^\mu(x)$ that depend on $x$, but this requires the introduction of a metric that transforms in the appropriate way.

We will now perform analogous arguments in the space of curves. Our arguments will necessarily be somewhat formal. We will consider a curve to be a periodic function $X^\mu(\lambda)$ and use the usual machinery of functional integration and differentiation over the space of periodic functions. Thus consider the following translation $\xi^\mu(X;\lambda)$ on the space of curves:

$$X^\mu(\lambda) \rightarrow X^\mu(\lambda) + \xi^\mu(X;\lambda). \tag{B.163}$$

Here the notation indicates that $\xi^\mu(X;\lambda)$ itself depends on the curve $X$ it is acting on. We now consider only the space of transformations that leave invariant the proper length element $\dot{X}^2$, as we need the reparam-invariant infinitesimal $d\lambda\sqrt{\dot{X}^2}$ in order to perform the integral over the kinetic term in the string field action (30). We impose:

$$\dot{X}^\mu \dot{\xi}_\mu(X;\lambda) = 0. \tag{B.164}$$

It is instructive to examine what a solution to the above condition looks like. We can construct such a solution in $d$ dimensions as

$$\xi^\mu(X;\lambda) = \int_0^\lambda d\lambda' \epsilon^{\mu\nu\alpha_1\cdots\alpha_{d-2}} \dot{X}_\nu(\lambda') V_{\alpha_1\alpha_2\cdots\alpha_{d-2}}(\lambda'), \tag{B.165}$$

where $V_{\alpha_1\cdots\alpha_{d-2}}(\lambda')$ is an arbitrary $(d-2)$-form defined on the worldline (and can depend on the curve $X$)[12].

Now consider the transformation of the string field:

$$\psi[X] \rightarrow \psi[X] + \oint d\lambda \frac{\delta\psi[X]}{\delta X^\mu(\lambda)} \xi^\mu(X;\lambda). \tag{B.166}$$

We will demand that the string field action $S[\psi]$ be invariant under this transformation. Let us consider the analogue of (B.161) by examining a sample action of the form:

$$S[\psi] = \int [dX] h[X] \psi[X]. \tag{B.167}$$

Under the transformation above, we find that the action varies as

$$\delta_\xi S[\psi] = \int [dX] h[X] \oint d\lambda \frac{\delta\psi[X]}{\delta X^\mu(\lambda)} \xi^\mu(X;\lambda). \tag{B.168}$$

Now we use the following functional integration identity

$$\int [dX] \frac{\delta}{\delta X^\mu(\lambda)} \phi[X] = 0, \tag{B.169}$$

which holds for an arbitrary functional $\phi[X]$, and tells us that "total derivative" terms arising from the boundary of the integration domain vanish. Integrating by parts in the functional integral, we then find that

$$\delta_\xi S[\psi] = -\int [dX] \left( \psi[X] \oint d\lambda\, \xi^\mu(X;\lambda) \frac{\delta h[X]}{\delta X^\mu(\lambda)} + h[X]\psi[X] \oint d\lambda \frac{\delta\xi^\mu(X;\lambda)}{\delta X^\mu(\lambda)} \right). \tag{B.170}$$

---

[12]We thank Zhengdi Sun for pointing out that if $V$ is independent of $X$ and $X, V, \xi$ are periodic then only rigid translations are allowed.

The analogue of the last term did not arise in our field theory example above in (B.162), because there we could assume that the transformation $\xi^\mu$ was independent of $x$. Let us thus explicitly compute this derivative using the expression (B.165)

$$\frac{\delta\xi^\mu(X;\lambda)}{\delta X^\mu(\lambda')} = \int_0^\lambda d\lambda'' \epsilon^{\mu\nu\alpha_1\cdots\alpha_{d-2}}\left(\frac{d}{d\lambda''}\delta(\lambda''-\lambda')g_{\mu\nu}V_{\alpha_1\alpha_2\cdots\alpha_{d-2}}(\lambda'')\right.$$
$$\left. + \dot{X}_\nu(\lambda'')\frac{\delta}{\delta X^\mu(\lambda')}V_{\alpha_1\alpha_2\cdots\alpha_{d-2}}(\lambda'')\right), \qquad (B.171)$$

where the first term vanishes by antisymmetry. We can constrain the dependence of $V$ on $X$ so that the second term vanishes.

We thus conclude that invariance under this loop-space translation requires:

$$\oint d\lambda\, \xi^\mu(X;\lambda)\frac{\delta h[X]}{\delta X^\mu(\lambda)} = 0. \qquad (B.172)$$

If $\xi$ were unconstrained, this would mean that the coefficients in the action couldn't depend on the curve at all; given the constraints (B.164) together with reparametrization-invariance, this means the only terms that can appear in the action (or the measure) are those that depend only on the proper length of the curve $L[C]$.

## C  Saddle-point evaluation of density of areas

Here we compute the measure over areas $g(a)$ defined in (46):

$$g(a) = \int [dX]\exp(-mL[C])\,\delta(a - A[C]), \qquad (C.173)$$

where the path integral is taken over all contractible curves and $A[C]$ is the minimal area that fills in the curve. First we write

$$g(a) = \int [dC]\int \frac{d\omega}{2\pi}\exp(-mL[C] + i\omega(a - A[C])). \qquad (C.174)$$

Now in order to do a saddle-point evaluation, let's vary the action in the exponent with respect to the worldline trajectory and with respect to $\omega$. We find

$$-m\ddot{X}^\mu - i\omega n^\mu = 0, \qquad a = A[C], \qquad (C.175)$$

where $n^\mu$ is an outwards pointing normal vector and the derivatives are taken with respect to proper length along the path. If we choose $\omega$ to be the correct imaginary number, then this equation has a solution which is simply a closed circle in Euclidean space with radius $R$, where

$$\omega = -i\Omega, \qquad \dot{\theta}^2 = \frac{\Omega}{mR}. \qquad (C.176)$$

Imposing that the distance along the circle is $2\pi R$, we find that $\Omega = \frac{m}{R}$ and also that $R = \sqrt{\frac{a}{\pi}}$. Now we can put all of this back into the integral; the only thing that contributes is the worldline tension with $L = 2\pi R$, and so we find

$$g(a) \sim \left(-2m\sqrt{\pi a}\right), \qquad (C.177)$$

i.e. the density of states at area $a$ is suppressed by the length of a circle with area $a$.

Note that we have suppressed the integration over the centre of mass of the string, which does not affect the area; thus the measure factor $g(a)$ also contains a factor of the spacetime volume, as required by translational invariance.

# D Equations of motion for string field

In this Appendix, we explore some aspects of the string field equation of motion described in (32). First, we discuss some details in its derivation, which will require us to integrate by parts. The validity of this somewhat formal procedure is easier to see if we use the formal representation for the area derivative given in terms of regular functional derivatives given in [10, 12]:

$$\frac{\delta\phi[C]}{\delta\sigma^{\mu\nu}} = \lim_{\epsilon\to 0}\int_{-\epsilon}^{+\epsilon} d\tau\,\tau\,\frac{\delta^2\phi[C]}{\delta X^{\mu}(t+\frac{\tau}{2})\delta X^{\nu}(t-\frac{\tau}{2})}\,. \tag{D.178}$$

As we can integrate ordinary functional derivatives by parts within a functional integral $[dX]$, this implies that the area derivative can also be directly integrated by parts.

Using this, we can now write the quadratic part of the string field action (30) as

$$S[\psi] = \mathcal{N}\int [dX]\left(-\frac{1}{2}\oint_C ds\,\psi^\dagger[C]\frac{\delta}{\delta\sigma_{\mu\nu}(s)}\left(L[C]^{-1}e^{-mL[C]}\frac{\delta\psi[C]}{\delta\sigma^{\mu\nu}(s)}\right)\right.$$
$$\left. + re^{-mL[C]}\psi[C]^\dagger\psi[C]\right), \tag{D.179}$$

where we have made explicit the factor of proper length appearing in the measure. Varying this with respect to $\psi[C]$, we may now write down the equations of motion directly in loop space:

$$-\frac{1}{2}e^{mL[C]}\oint_C \frac{\delta}{\delta\sigma_{\mu\nu}(s)}\left(ds\,\frac{e^{-mL[C]}}{L[C]}\frac{\delta\psi[C]}{\delta\sigma^{\mu\nu}(s)}\right) + r\psi[C] = 0\,. \tag{D.180}$$

In (D.180) the action of the area derivative on most of the parts of the expression is straightforward to understand. There is a subtlety however associated with the action of the area derivative on $ds$, the integration measure over the curve itself. To understand this, it is helpful to switch to a general parameter $\lambda$, in terms of which we have $ds = d\lambda\sqrt{\dot{X}^2}$. This part of the expression can now be written as

$$\frac{\delta}{\delta\sigma_{\mu\nu}(\lambda)}\sqrt{\dot{X}^2(\lambda)} = \lim_{\lambda'\to\lambda}\frac{\delta}{\delta\sigma_{\mu\nu}(\lambda')}\sqrt{\dot{X}^2(\lambda)} = -\lim_{\lambda'\to\lambda}\frac{2}{\sqrt{\dot{X}^2}}\dot{t}_{[\mu}t_{\alpha]}\delta(\lambda-\lambda')\,, \tag{D.181}$$

where we have used (A.151). The short distance divergence seems to appear from the fact that we integrate the point where we compute the area derivative over the whole curve; as we will see, this results in an effective renormalization of the worldline tension $m$. Returning to the proper length parametrization $s$, we denote $\delta(s-s'=0) = \Lambda$.

The remainder of the expression is more straightforward. Let us now compare these with the equations of motion previously obtained in Section 3.2. We consider again the restricted form $\psi[C] = f(A[C])$. Plugging in and using the expressions for area derivative and length derivatives in (A.150), we find

$$-\left(\frac{1}{L[C]}\left(L[C]^{-1}+m+\Lambda\right)\oint ds\,t^\nu n_\nu\right)f'(a) - f''(a) + rf(a) = 0\,. \tag{D.182}$$

We now see an issue with this ansatz; the first term in $f'(a)$ depends the integral of the tangent and normal vectors, which is not in general a function of the area of the curve. Thus it is not possible to satisfy this equation; however as argued in the bulk of the paper it is possible to find a solution for asymptotically large areas. It would be very interesting to solve this equation more generally.

# E  Computation of kinetic term for scalar phase modulation

Here we continue to compute the kinetic term for the field $t(x)$ arising from evaluating the string field action on the scalar phase modulation ansatz (78). As details have been explained in the bulk of the text, we will here be somewhat brief. The area derivative is

$$\frac{\delta \psi_t[C]}{\delta \sigma^{\mu\alpha}(s)} = 2i \left( \frac{\dot{X}_{[\mu}\partial_{\alpha]}t(X(s))}{m\epsilon} - \frac{t(X(s))\ddot{X}_{[\mu}\dot{X}_{\alpha]}}{(m\epsilon)^3} \right) \psi_t[C]. \tag{E.183}$$

This is then squared and path-integrated over in the action. The contribution arising from the square of the second term has already been worked out in the bulk of the paper; we must thus compute the square of the first term. The cross term involves an odd number of $X$'s and so will vanish.

The square of the first term is

$$\frac{4}{(m\epsilon)^2} \left( \dot{X}_{[\mu}\partial_{\alpha]}t\dot{X}^{[\mu}\partial^{\alpha]}t \right) = (\partial_\alpha t \partial^\alpha t) - \frac{2}{(m\epsilon)^2}(\dot{X}^\alpha \partial_\alpha t)^2. \tag{E.184}$$

The path integral over the first term involves no new kinematic dependence on the worldline and so is essentially exactly the same as the computation of the gauge field kinetic term in (77). To determine the second term we need to work slightly harder. Define the Fourier transform of the function $T_{\alpha\beta}(x) = v^2 \partial_\alpha t(x)\partial_\beta t(x)$:

$$T_{\alpha\beta}(x) = \int \frac{d^d k}{(2\pi)^d} e^{+ik\cdot x} \tilde{T}_{\alpha\beta}(k), \tag{E.185}$$

and then the answer for the contribution to the action is $\int \frac{d^d k}{(2\pi)^d} K_{\alpha\beta}(k)\tilde{T}^{\alpha\beta}(k)$, where the form factor is:

$$K^{\alpha\beta}(k) \equiv -\frac{2\mathcal{N}}{N(m\epsilon)^2}(2\pi)^d \delta^{(d)}(k) \int \frac{dL}{L^{1+\frac{d}{2}}} e^{-\frac{m^2 L^2}{2}} \oint \frac{ds'}{2L} \langle \dot{X}^\alpha(s')\dot{X}^\beta(s') \rangle. \tag{E.186}$$

We now explicitly compute from the definition of the worldline propagator

$$\langle \dot{X}^\alpha(s')\dot{X}^\beta(s') \rangle = -\lim_{s \to s'} \epsilon \left( \delta(s-s') - \frac{1}{L} \right) \delta^{\alpha\beta}. \tag{E.187}$$

As before, we need to regulate the delta function; using the same representation (84) we have $\delta(0) \to \frac{1}{\sqrt{2\pi\Delta}}$. As $\Delta$ is the cutoff we always have $\Delta \ll L$ and can ignore the $L$ dependence in the integral. Absorbing the $L$ integral into the definition of $\mathcal{N}$, we find then that the form factor is

$$K^{\alpha\beta}(k) = (2\pi)^d \delta^{(d)}(k) \frac{1}{\sqrt{2\pi\Delta}\epsilon m^2} \delta^{\alpha\beta}. \tag{E.188}$$

Assembling the pieces, we find the following for the kinetic term:

$$S_{\text{kinetic}}[t] = v^2 \int d^d x \, (\partial t)^2 \left( \frac{1}{2} + \frac{1}{\sqrt{2\pi\Delta}\epsilon m^2} \right). \tag{E.189}$$

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
