# Peer review of "Mean string field theory: Landau-Ginzburg theory for 1-form symmetries"

_SciPost Physics, doi:SciPost Phys. 13, 114 (2022)_

## Round 1 · Author Response

We thank both of the referees for their careful reading and for the positive and helpful comments. The first referee did not ask for any clarifications. The second referee suggested that we clarify some points further, and we have thus made some minor comments to the draft. Our responses are below. We have included the a copy of the referee questions, placed inside "//" symbols to set them apart from the text.

//
So, regarding this derivative operator, is it clear that the definition they use is the correct/physical/best/unique/… one? While the authors have an extensive discussion of the derivative, the answers to these questions are still not obvious to me. //

To the best of our knowledge, any other derivative cannot be made gauge-covariant with respect to an external 2-form source. This is an important physical point, and we have now further emphasized it below Eq (2.20) with the words:

“Note that the area derivative is precisely the correct object for this construction, and indeed this is why the formalism requires the area derivative rather than a more conventional functional derivative with respect to the coordinates of the curve $X(\lambda)$”

(as well as a footnote providing some more background).

//
Is it clear that one can neglect higher derivative terms? Is there some reason that they are suppressed in the IR in a similar way to normal derivative terms in standard EFTs?
//

So the usual derivative expansion about a free field theory is basically just dimensional analysis. (Subtleties that arise when there are marginal terms or when one tries to expand about a strongly-coupled fixed point can be ignored here.) Dimensional analysis also applies here, so we can attempt to apply the same reasoning, and thus the answer to both questions of the referee is expected to be “Yes”. We have made this somewhat more explicit in the words following Eq (7.4), which now read:

“(Compare with [φ] = d−2 for field theory; the difference arises from the dimension of the area derivative, which is 2). We can now attempt to use dimensional analysis to suppress higher derivative terms as usual in conventional field theory”.

(As mentioned in the text, however, it is somewhat difficult to apply the reasoning to topology-changing terms).

//
One other question: above eq. (2.9), what does it mean that the upper critical dimension for U(1) is 4? Why is it related to the group choice and what is it for other groups? It also seems that there is a word (maybe “for”) missing in “(which the U(1) case above is 4)”
//

We thank the referee for pointing out the typo, which we have now fixed. We define (as usual) the upper critical dimension for the theory to be the dimension where the leading interaction coupling becomes marginal. We specify the group $U(1)$ simply because that's the case that has a U(1) 1-form symmetry, which is the example we focus on. With other choices of gauge group, the 1-form symmetry will be different and the mean string field theory must be modified accordingly; in particular the leading interaction and the corresponding upper critical dimension for the theory might change.

//
Finally, would it be useful to describe in greater detail why one cannot write down a mean field theory for open strings? Surely, some condensed matter system can be made of open dynamical topological defects.
//

Here we should re-emphasize that our starting point is not a theory of strings, but a theory of linearly-realized 1-form symmetry. The fact that this involves (closed) strings is an output of the analysis; we have made this more explicit below Eq (2.10), where we state

“Note that the curves C must be closed loops for invariance under the 1-form symmetry, and thus our framework only involves closed strings.”

We are not aware of a clean symmetry principle that would produce a theory of open strings; in some examples they appear to arise when a 1-form symmetry is explicitly broken at some high scale which determines the worldline tension of the “edge” of a string; if this scale can be parametrically separated from the string tension the system may admit a useful description in terms of open strings. It would be very interesting to try and use our framework to study such a system, but it it outside the scope of the current work.

---

## Round 1 · List of Changes

See above.

You are currently on this page

Resubmission scipost_202209_00052v1 on 24 September 2022

---

## Editorial Decision

published